# Structure of the *M. tuberculosis* DnaK–GrpE complex reveals how key DnaK roles are controlled

Xiansha Xiao[1], Allison Fay [2], Pablo Santos Molina[2], Amanda Kovach [1], Michael S. Glickman [2] & Huilin Li [1] ✉

The molecular chaperone DnaK is essential for viability of *Mycobacterium tuberculosis* (Mtb). DnaK hydrolyzes ATP to fold substrates, and the resulting ADP is exchanged for ATP by the nucleotide exchange factor GrpE. It has been unclear how GrpE couples DnaK's nucleotide exchange with substrate release. Here we report a cryo-EM analysis of GrpE bound to an intact Mtb DnaK, revealing an asymmetric 1:2 DnaK–GrpE complex. The GrpE dimer ratchets to modulate both DnaK nucleotide-binding domain and the substrate-binding domain. We further show that the disordered GrpE N-terminus is critical for substrate release, and that the DnaK–GrpE interface is essential for protein folding activity both in vitro and in vivo. Therefore, the Mtb GrpE dimer allosterically regulates DnaK to concomitantly release ADP in the nucleotide-binding domain and substrate peptide in the substrate-binding domain.

The 70-kDa heat shock protein (Hsp70 in eukaryotes and DnaK in prokaryotes) is a molecular chaperone found in all domains of life as well as in most of the subcellular components of eukaryotic cells[1–3]. DnaK functions in a variety of cellular processes, including protein folding, prevention of aggregation, protein transport across membranes, and protein degradation[4–6]. DnaK is assisted by DnaJ (Hsp40 in eukaryotes, a J-domain containing cochaperone) and GrpE, a nucleotide exchange factor (NEF) (Fig. 1a). DnaK is composed of two main domains, the N-terminal nucleotide-binding domain (NBD) and the C-terminal polypeptide substrate-binding domain (SBD), and these domains are connected via a highly conserved but flexible linker (Fig. 1b)[7].

Central to the function of DnaK is the intricate allosteric regulation between NBD and SBD[8,9]. In the ATP-bound state, DnaK's affinity for substrate is low but substrate association and disassociation rates are high. Upon ATP hydrolysis, affinity for substrate increases 10- to 50-fold, but substrate association and dissociation rates decrease 100- and 1000-fold, respectively[10,11]. The structures of the ATP- and ADP-bound DnaK rationalized these differences. In the ADP-bound or nucleotide-free states, the two functional domains have little interaction[7,8,12,13], but upon ATP binding, the two domains are tightly coupled, allowing drastically accelerated kinetics in both binding and release of polypeptide substrate[10,14,15]. This ATP-induced allosteric coupling is crucial for efficient chaperone activity[9]. However, the intrinsic ATP hydrolysis rate is low and requires timely stimulation by a DnaJ bound protein substrate[16,17]. While DnaJ accelerates ATP hydrolysis by DnaK, their release of tightly bound ADP is facilitated by the NEF GrpE, which in turn promotes substrate release[18,19].

In the *E. coli* system, DnaJ recruits a substrate protein to DnaK in the ATP-binding state in which the SBD is in an open conformation (Fig. 1a)[19,20]. Hydrolysis of ATP by DnaK generates the ADP state in which the α-helical lid is closed to facilitate substrate folding[14,15,22]. Subsequent binding of GrpE to DnaK promotes the exchange of ADP for ATP, opening of the SBD to allow substrate release[20]. The ATP-hydrolysis-dependent release of substrate from DnaK is stimulated by GrpE in a manner that is analogous to the regulation of many GTP-binding proteins[18]. In a landmark crystallographic study, the *E. coli* GrpE was found to assemble a dimer that bound asymmetrically to a single copy of the DnaK NBD, and that the NBD was held by the GrpE β-sheet domain in an open conformation that was incompatible with nucleotide binding, explaining the GrpE-mediated nucleotide exchange mechanism in DnaK[23]. However, because only the NBD was

[1]Department of Structural Biology, Van Andel Institute, Grand Rapids, MI, USA. [2]Immunology Program, Sloan Kettering Institute, New York, NY, USA. ✉e-mail: Huilin.Li@vai.org

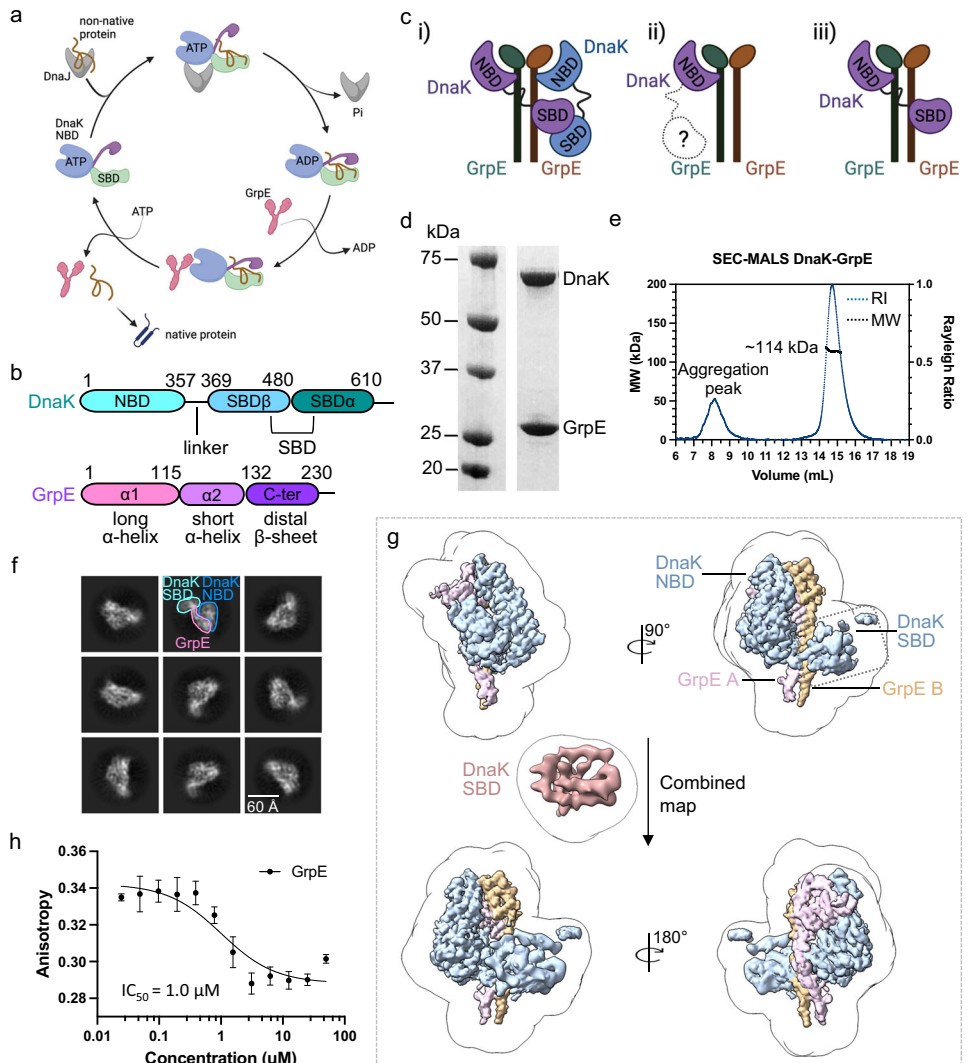

**Fig. 1 | Functional cycle of the prokaryotic DnaK/DnaJ/GrpE system and cryo-EM analysis of the Mtb DnaK–GrpE complex. a** Schematic representation of the DnaK/DnaJ/GrpE reaction cycle. This figure is modified from ref. 66. **b** Domain architectures of the Mtb DnaK and GrpE. **c** Three possible binding modes between DnaK and the GrpE dimer. Left panel is based on the crystal structure of the *G. kaustophilus* GrpE–DnaK complex (PDB ID 4ANI). Middle panel is based on the crystal structure of the *E. coli* GrpE dimer bound to the DnaK NBD (PDB ID 1DKG). Right panel is from current study of the Mtb system. **d** SDS-PAGE gel of the in vitro assembled Mtb DnaK–GrpE complex, which was done once. **e** The DnaK–GrpE peak was estimated to have a mass of 114 kDa by SEC-MALS analysis, consistent with a stoichiometry of 1:2 DnaK/GrpE with a theoretical mass of 115 kDa. **f** Selected 2D class averages of the Mtb DnaK–GrpE complex. **g** EM map of the Mtb DnaK–GrpE complex. The map is surface rendered at 4σ and colored by subunits. A low pass filtered gray mask is superimposed to show the overall contour of the whole complex. In the middle panel, the focus refined DnaK SBD is shown separately in salmon. **h** Fluorescence polarization of the F-NR bound DnaK at an increasing concentration of GrpE. GrpE stimulates the release of F-NR from DnaK with a rough half-maximal inhibition concentration of 1.0 μM. *n* = 3 biologically independent samples. Data are presented as mean values +/− SD. Source data are provided as a Source data file. Panels (**a**) and (**c**) was created with BioRender.com.

used in that study, it has been unclear if the 1:2 stoichiometry holds true for the full-length proteins, and how GrpE interacts with the DnaK SBD. Confusion about the exact GrpE:DnaK stoichiometry was further exacerbated by another crystallographic study in which the full-length DnaK and GrpE from *Geobacillus kaustophilus* were found to form a 2:2 complex in which the GrpE dimer bound to two copies of DnaK (Fig. 1c)[24].

To clarify the interaction, we analyzed the DnaK–GrpE system from *Mycobacterium tuberculosis* (Mtb) by cryo-EM, an approach that circumvents the requirement of crystal lattice formation which is often a source of artefactual protein-protein interactions. DnaK is nonessential in normal growth conditions in *E. coli* but is essential in mycobacteria for cell growth and protein folding[25]. It plays critical roles in Mtb physiology and persistence in the host[26,27]. Therefore, the Mtb DnaK is a potential target for development of anti-TB therapeutics[28]. We show that the full-length DnaK binds GrpE in 1:2 stoichiometry and confirm

that the observed interfaces between GrpE and DnaK are essential for substrate protein refolding in vitro and in vivo. We identify key sites in the GrpE that regulate substrate release from DnaK SBD. Our work reveals that the GrpE regulates DnaK for concomitant nucleotide exchange in the NBD and peptide release in the SBD.

## Results

### In vitro reconstitution and cryo-EM analysis of the Mtb DnaK system

Mtb contains two DnaJ proteins, with *dnaJ1* encoded in the *dnaK* operon and *dnaJ2* encoded elsewhere in the genome, which are highly conserved[28]. Because either DnaJ1 or DnaJ2 can support Mtb viability and DnaK activity[28], we used DnaJ2 in this study. We reconstituted the ternary substrate refolding DnaK–DnaJ2–GrpE system in vitro by mixing the three separately purified proteins at a molar ratio of 1:2:2 and in the presence or absence of nucleotide (ADP or ATP). Size

exclusion chromatography (SEC) of the mixtures revealed the formation of a stable complex composed of DnaK and GrpE (Fig. 1d). The complex had an estimated mass of 114 kDa based on a SEC-MALS (size exclusion chromatography coupled with multi-angle light scattering) analysis, suggesting a complex with a stoichiometry of 2 GrpE: 1 DnaK, which has a calculated mass of 115 kDa (Fig. 1e).

The SEC peak fractions were then vitrified for cryo-EM analysis. Two-dimensional classification and averaging of the particle images showed densities of DnaK and GrpE, but not DnaJ2 (Fig. 1f), indicating that DnaJ2 is highly flexible on DnaK−GrpE. Consistent with these data, the Mtb DnaJ2 was not observed in a previous study of the DnaJ2-bound ClpB-DnaK bi-chaperone system[29]. Our cryo-EM analysis showed that 38% of DnaK complexed with GrpE in the apo state, but the binary DnaK−GrpE complex did not form in the presence of ATP (Supplementary Fig. 1a, b). We went on to derive a cryo-EM map of the DnaK−GrpE complex at an average resolution of 3.7 Å (Supplementary Figs. 2 and 3). The α-lid of the DnaK SBD was flexible and of low resolution in this map (Fig. 1g). Therefore, we performed local refinement on SBD, yielding an improved EM map for this region at 5.8 Å (Fig. 1g). The SBD map was then combined with the overall map to generate a composite 3D map, which allowed atomic modeling in most regions except for the SBD lid. We used the AlphaFold2 predicted structural model to fit the SBD density. Only the first two α-helices in SBDα had EM density, while the remaining three short α-helices had no density and were not modeled. Further, the GrpE N-terminal 44 residues and C-terminal 48 residues (aa 188–235) were disordered thus not modeled. The β-sheet domain of GrpE molecule A was stabilized by DnaK NBD, but the β-sheet domain of GrpE molecule B was disordered and was not modeled. Addition of a substrate peptide, NRLLLTG (NR), or a peptide inhibitor, Telaprevir[30], did not stabilize the SBD in the complex (Supplementary Fig. 1c, d). Therefore, the DnaK−GrpE complex is likely in a post-reaction state in which both ADP and substrate have been released and the SBD lid becomes partially flexible.

We next investigated if the added nucleotides (ADP or ATP), the substrate peptide, or the peptide inhibitor remained bound to the in vitro assembled DnaK−GrpE complex. We first removed any unbound nucleotides or substrate/inhibitor by gel filtration. Mass spectrometry of the peak fraction corresponding to the DnaK−GrpE complex detected neither nucleotides nor substrate peptide/inhibitor in the complex (Supplementary Fig. 4). These findings support our suggestion that the DnaK−GrpE complex is in a state in which both ADP and substrate have been released.

We further investigated whether GrpE-stimulated substrate release from DnaK. We used a fluorescently tagged NR (F-NR) and performed a fluorescence polarization (FP) experiment. We first confirmed that F-NR bound to the Mtb DnaK and showed that the F-NR binding affinity for DnaK was high, and the binding kinetics was slow in the presence of ADP or in the absence of any nucleotide (Supplementary Fig. 5). In the competition assay, we pre-incubated DnaK with ADP and F-NR before adding an increasing concentration of GrpE protein (Fig. 1h). We found that GrpE competed with F-NR for binding to DnaK with an estimated half-maximal inhibition concentration ($IC_{50}$) of 1.0 μM, indicating that GrpE efficiently stimulates substrate release from DnaK. This result further supports our structure-based suggestion (described below) that the captured DnaK−GrpE complex is in a state in which both ADP and substrate have been released.

## Asymmetric binding between DnaK and GrpE

The structure of the intact complex reveals that one molecule of DnaK binds asymmetrically to a GrpE dimer, with DnaK NBD and SBD straddling across the GrpE dimer (Fig. 2a). Therefore, the stoichiometry of the Mtb GrpE and DnaK is 2:1, consistent with the above-described 2:1 ratio by solution SEC-MALS analysis (Fig. 1e). A previous crystal structure of a full-length *G. kaustophilus* GrpE−DnaK complex suggested a stoichiometry of 2:2[24], but in another crystallographic

study of the *E. coli* DnaK NBD−GrpE complex, the stoichiometry was 2:1[23]. The prokaryotic GrpE−DnaK system is well conserved with high sequence identity (Fig. 2a, Supplementary Fig. 6a)[31]. Superimposition of the DnaK−GrpE structures of Mtb, *E. coli*, and *G. kaustophilus* revealed an overall r.m.s.d. value of 3.9 Å between Mtb and *E. coli* structures, and 5.3 Å between Mtb and *G. kaustophilus* complexes, suggesting a conserved overall structure with significant conformational differences (Supplementary Fig. 7). We suggest that all prokaryotic GrpE and DnaK systems likely interact with the 2:1 stoichiometry.

As expected, the Mtb GrpE structure is composed of a long N-terminal α-helix, a central four-helix bundle, and a C-terminal β-sheet (Figs. 1b, 2a). The Mtb DnaK resembles a canonical Hsp70 protein composed of the N-terminal NBD and the C-terminal SBD (Fig. 2a). The DnaK NBD can be divided into two lobes (I and II) with each lobe further divided into two subdomains (A and B): subdomain IB and IIB form the composite nucleotide-binding pocket; and subdomain IA and IIA have a similar topology, are related by a pseudo twofold symmetry axis, and form the base of the nucleotide-binding cleft. The SBD is divided into the β-sheet subdomain (SBDβ), which contains peptide binding pocket and the α-helical subdomain (SBDα) that functions as a lid (α-lid).

The Mtb DnaK adopts a disjoined conformation in which the NBD and SBD are separated with no direct interactions, and the α-lid is in a closed state. Therefore, the Mtb GrpE bound DnaK resembles the ADP or nucleotide-free state[12] and is different from the ATP state in which the α-lid stretches across NBD enabling SBDβ to directly interact with NBD[14]. Interestingly, the disjoined SBD domains exhibit diverse orientations with respect to the NBD as seen in the structures of *Mycoplasma genitalium* DnaK in the ADP state, *E. coli* DnaK in apo state, and Mtb DnaK (Supplementary Fig. 6b). The substantial difference in relative position and orientation of SBD must have been enabled by the flexible linker between the two domains.

## The interaction between GrpE and DnaK involves three contact regions

There are three significant contact regions between DnaK and the GrpE dimer (Fig. 2b, c). The major contact (contact region I) is between the proximal β-sheet domain of the GrpE A and the DnaK NBD subdomains IB and IIB. The other two interfaces are between the long N-terminal α-helix of the GrpE A and the DnaK NBD subdomains IA and IIA (contact region II) and between the long N-terminal α-helix of the GrpE B and DnaK SBD (contact region III). The interaction between DnaK NBD and GrpE A buries 1417 Å² solvent accessible surface, and the interaction between DnaK SBD and GrpE B buries 891 Å² solvent accessible surface.

The full-length complex structure reveals numerous charge-charge interactions between DnaK and the GrpE dimer. In contact region I, the DnaK NBD subdomain IB interacts with the face of the proximal β-sheet domain of the GrpE molecule A (Fig. 3a–c). The DnaK Arg44 H-bonds with GrpE Ser160, the DnaK Asn56 H-bonds with the main chain carbonyl oxygen of the GrpE Pro144, and the DnaK Thr60 H-bonds with the main chain carbonyl oxygen of the GrpE His152 (Fig. 3a). The DnaK NBD subdomain IIB interacts with the two short α-helices of the GrpE dimer by forming a salt bridge between DnaK Glu236 and GrpE molecule A Arg169, an H-bond between DnaK Tyr257 and GrpE molecule A Asp110, and another H-bond between DnaK Tyr257 and the GrpE B Ser118 (Fig. 3b).

In contact regions II and III, the long N-terminal α-helices of the GrpE dimer harbor a series of positively charged residues (Arg64-Arg86) that form extensive interactions with the DnaK NBD subdomain IA and SBD, respectively (Fig. 3c–e). At the interface between DnaK IA and the long α-helix of GrpE molecule A, the DnaK Tyr106 H-bonds with the GrpE Arg78, while DnaK Glu20 and Glu359 form salt bridges with the GrpE molecule B Arg73 and GrpE molecule A Arg64, respectively (Fig. 3d). The interface is further stabilized by two additional H-bonds between the DnaK Gly32 main chain carbonyl oxygen and GrpE Arg86. Notably, the long α-helix of the GrpE molecule B also

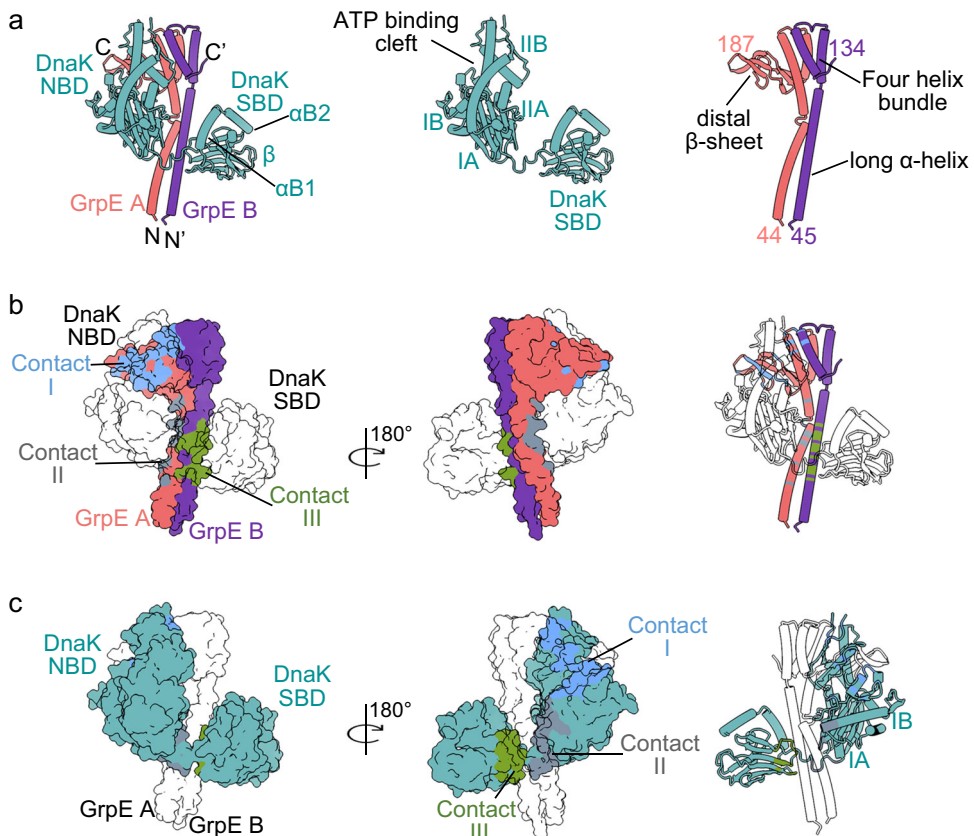

**Fig. 2 | Structure and interaction of the Mtb DnaK–GrpE complex. a** Cartoon view of the Mtb DnaK–GrpE structure and individual subunits shown separately. N, N terminus; C, C-terminus. **b** Surface view of the DnaK–GrpE structure showing the DnaK footprint on the GrpE dimer. DnaK is transparent white. Blue and gray show the DnaK contact areas (contact regions I and II) on the orange GrpE molecule A. Green shows the DnaK contact areas (contact region III) on the purple GrpE molecule B. **c** Surface view of the DnaK–GrpE structure showing the GrpE footprint on DnaK. The GrpE dimer is transparent white. Blue and gray show the contact area of the GrpE molecule A on DnaK and green the contact of the GrpE molecule B on DnaK.

contributes to binding to the DnaK SBD (Fig. 3e). The GrpE Arg64 forms an H-bond with the main chain carbonyl oxygen of Thr394 and a salt bridge with Asp451 of DnaK. Finally, the GrpE Lys74 H-bonds with the DnaK Thr390 and with main chain atoms of Arg388 and Asn389, respectively. Interestingly, Arg64 and Arg75 of the GrpE long N-terminal α-helices mediate the interaction with both DnaK NBD and SBD domains.

## The long N-terminal α-helix of GrpE is essential for protein refolding by DnaK

To examine the functional importance of the DnaK–GrpE interaction, we made four sets of mutations in GrpE and four sets of mutations in DnaK and assayed for their ability to reactivate the denatured luciferase. In GrpE, the four patches of mutations are (Fig. 4a, Supplementary Fig. 6a): (i) Mut-1 includes residues in the long N-terminal α-helix (R64A, N71A, R73A, K74A, R75A, and R78A) which interact with either DnaK NBD subdomain IA in contact region II or DnaK SBD in contact region III; (ii) Mut-2 includes residues in three short α-helices that contact DnaK NBD subdomain IIB (D110A and P115A) in contact region I; (iii) Mut-3 includes residues in the proximal β-sheet region which interact with DnaK NBD subdomain IIB (S160A, R169A and H181A) in contact region I; and (iv) Mut-4 includes residues in the long N-terminal α-helices (Arg64A and Arg75A) that interact with both DnaK NBD subdomain IIB and DnaK SBD.

We purified these mutant proteins and examined the luciferase refolding activity in the presence of DnaJ2 and 2 mM ATP. As expected, the WT DnaK and GrpE reactivated denatured luciferase (Fig. 4b), though incompletely compared with the undenatured luc. The in vitro

refolding procedure we used had been optimized in a previously study, and we achieved a comparable luciferase recovery percentage[28]. Briefly, firefly luciferase was denatured at 42 °C and refolded by DnaK, DnaJ2, and GrpE in 2 mM ATP at 25 °C. We found that the GrpE Mut-3 had no effect, and the GrpE Mut-2 had a modest reduction to the luciferase reactivation by the WT DnaK (Fig. 4b). The G122D mutation of the *E. coli* GrpE conferred a temperature-sensitive growth phenotype[19], but this residue is absent in Mtb GrpE. In contrast, the GrpE Mut-1 and Mut-4 greatly reduced the luciferase reactivation (Fig. 4b). A previous mutagenesis screen demonstrated that a mutation at Glu53 of the *E. coli* GrpE, equivalent to the Mtb GrpE Glu57, gave rise to a temperature sensitive phenotype for λ replication[32]. Because Mut-1, -4 and Glu57 are all located in the GrpE long N-terminal α-helix that mediates the interactions to DnaK NBD IA domain and DnaK SBD domain, these results indicate that the interaction of the GrpE long N-terminal α-helix domain to either DnaK NBD IA domain or DnaK SBD domain are essential for the in vitro function of the DnaK chaperone system. Intriguingly, residues from Mut-1 and Mut-4 region (aa 60–80) are highly conserved in the GrpE family, whereas the ones from Mut2 and Mut3 regions are less conserved (Supplementary Fig. 6a). We also examined GrpE mutations on the ADP/ATP exchange rate of DnaK (Fig. 4c). Consistent with the luciferase refolding activity, GrpE Mut-1 and Mut-4 nearly abolished the ADP/ATP exchange. GrpE Mut-2 also significantly reduced the ADP/ATP exchange rate, but GrpE Mut-3 had little effect. We hypothesized that the greatly compromised refolding activity and nucleotide exchange rate by GrpE Mut-1 were due to its diminished binding to DnaK. Isothermal titration calorimetry (ITC) analysis showed that GrpE containing Mut-2 and Mut-3 mutations

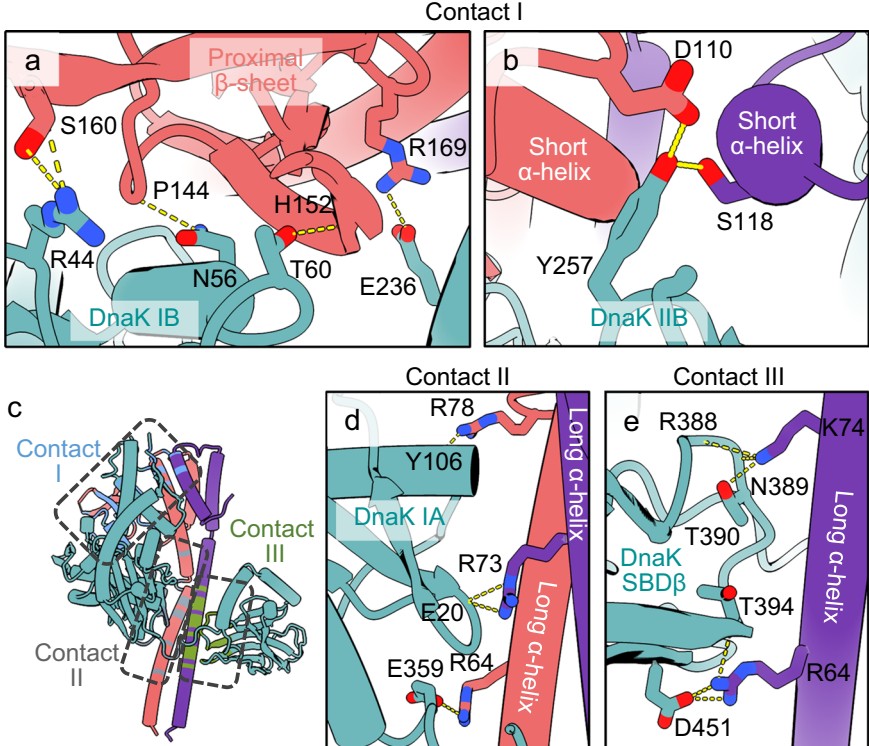

**Fig. 3 | Detailed interactions between the Mtb DnaK and the GrpE dimer.**
**a**, **b** Close-up views of the interacting residues between DnaK and GrpE in contact region I. The DnaK NBD subdomains IB and IIB interact with the face of the proximal β-sheet domain of the GrpE dimer. **c** The three contact regions detailed in other panels are mapped on the DnaK–GrpE structure and enclosed in the dashed rectangles. **d**, **e** In contact regions II (**d**) and III (**e**), the long N-terminal α-helix of the GrpE molecules A and B harbor a series of positively charged residues (Arg64-Arg86) that form extensive interactions with the DnaK NBD subdomain IA and the SBD. The interface residues are shown in sticks. Dashed lines indicate hydrogen bonds or salt bridges.

bound to DnaK with similar affinities as the WT GrpE, but GrpE Mut-1 had greatly reduced affinity for DnaK (Fig. 4d, panels i–iv). Consistent with the functional importance, the GrpE Mut-1 region is highly conserved (Fig. 4e).

We next asked whether the structurally defined contacts between GrpE and DnaK that affect protein refolding in vitro are also essential in vivo. We introduced the three *M. tuberculosis* GrpE mutants (Mut-1, Mut-2, and Mut-3) into *M. smegmatis* (Ms) and assessed the ability of each to complement for the essential function of GrpE in vivo. We utilized a strain of *M. smegmatis* with a chromosomal deletion of *grpE* and a second copy provided at the *attB*(L5) site marked with kanamycin resistance. We then tested for the ability to swap this second copy of Ms *grpE* for the entire Mtb DnaK operon with wild type or mutated ALFA epitope tagged *grpE* alleles encoded on integrating vectors marked with streptomycin resistance[33,34]. WT Mtb *ALFA-grpE* replaced Ms *grpE* with high efficiency (>2 × 10$^5$ streptomycin-resistance colonies per transformation) confirming that Mtb *grpE* tagged with ALFA is functional (Fig. 4f). Furthermore, both Mtb GrpE Mut-2 and Mut-3 allele supported viability, demonstrating that these mutant proteins could complement for essential GrpE function in *M. smegmatis* (Supplementary Fig. 8a). However, GrpE Mut-1 (Fig. 4f) or GrpE Mut1-3 (Supplementary Fig. 8a) did not support viability, indicating that the Mut-1 residues are essential for DnaK system.

To confirm these results, we utilized a system in which the endogenous *M. smegmatis dnaK* operon can be depleted using an inducible CRISPRi[35], which is lethal due to the essential function of DnaK (Fig. 4g). We then complemented this depletion strain with the Mtb DnaK operon, which is resistant to CRISPRi depletion, to test the function of mutant DnaK and GrpE alleles. As expected, strains containing wild-type Mtb *dnaK* operon were rescued from the lethality of endogenous DnaK depletion. However, although an Mtb *dnaK* operon

with *grpE* alleles with Mut-2 or 3 patch mutations supported viability, Mut-1 mutations did not, confirming that the Mut-1 patch was required for essential function (Fig. 4g). We confirmed in vivo expression of mutated Mtb GrpE proteins utilizing the ALFA tag to differentiate Mtb GrpE from Ms GrpE and found that all ALFA tagged Mtb GrpEs were expressed at similar levels and migrated near predicted size (Fig. 4h), indicating that protein expression level does not explain the in vivo dysfunction of GrpE Mut-1. Immunoprecipitation of WT Mtb GrpE-ALFA protein co-immunoprecipitated DnaK, as did GrpE Mut-2, and Mut-3 GrpE (Fig. 4i). However, Mut-1 and Mut-1, 2, 3 GrpE-ALFAs did not co-immunoprecipitate DnaK protein above the level observed in negative control pulldowns (Fig. 4i, j) indicating that the Mut-1 patch was required for DnaK–GrpE interaction in vivo.

Overexpression of GrpE in mycobacteria inhibits DnaK function and induces the DnaK operon through relief of HspR repression[25,36]. To confirm the functional inactivation of the GrpE mutants that are unable to interact with DnaK, we used a DnaK operon mCitrine promoter-reporter to assay the effect of GrpE allele overexpression on DnaK function. Overexpression of WT untagged or ALFA tagged GrpE induced the reporter, but GrpE with mutation patches 1–3 or 1 alone did not, confirming that GrpE Mut-1 is nonfunctional (Fig. 4k). These results demonstrate the physiological relevance of the long N-terminal α-helix of GrpE in mediating protein refolding by the Mtb DnaK.

## The DnaK surface interacting with the GrpE proximal β-sheet domain is important both in vitro and in vivo

We next investigated the functional importance of GrpE-interacting residues of DnaK by generating four cluster mutations (Mut1-4) in DnaK (Fig. 5a, Supplementary Fig. 9): Mut-1 in the N-terminal α-β fold of NBD IA (Y106A and L107A), Mut-2 in the DnaK NBD IIB subdomain (E236A and Y257A), Mut-3 in the distal α-helix of the NBD IA (E359A and

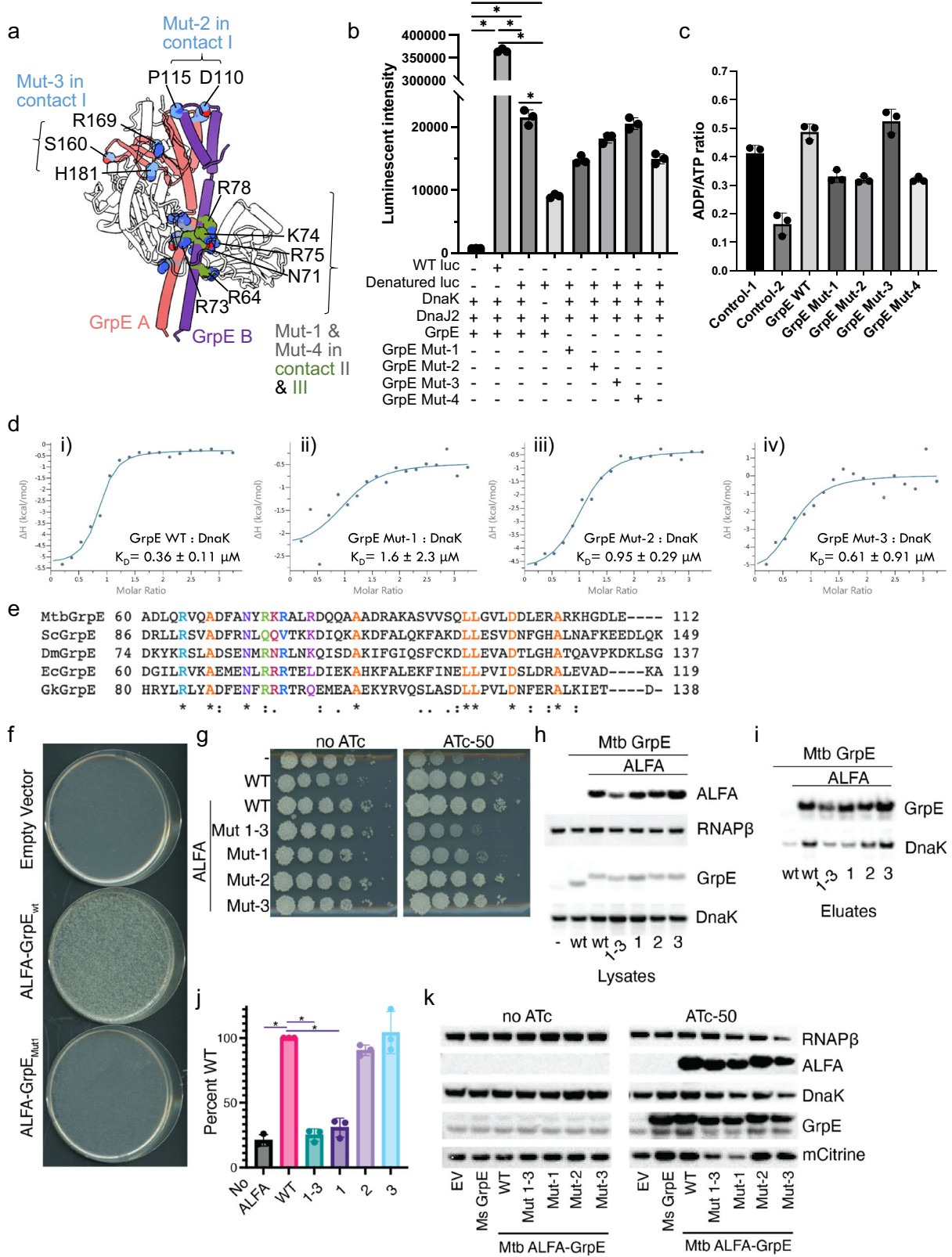

D362A), and Mut-4 in the DnaK SBD (E387A, T390A, T394A, and K395A). We purified DnaK proteins carrying the Mut-1, Mut-2, Mut-3, Mut-4. DnaK's ability to reactivate luciferase was nearly completely blocked by Mut-2 mutation, greatly reduced by Mut-1 and Mut-4, and modestly blocked by Mut-3 mutation (Fig. 5b). Mtb DnaK Mut-3 mutation (Glu359 and Asp362) is equivalent to Asp354 and Asp357 in

the *G. kaustophilus* DnaK, which was shown by in vivo complementary assays to be essential for cell viability after heat shock[24]. We examined whether DnaK mutations (Mut-1, Mut-2, Mut-3, and Mut-4) affect the binding affinity to GrpE by ITC and found that DnaK Mut-2 abolished the interaction with GrpE, whereas DnaK Mut-1, Mut-3, and Mut-4 showed similar binding affinity to GrpE as WT DnaK (Fig. 5c),

**Fig. 4 | GrpE mutations affect DnaK function and interaction with DnaK in vitro and in vivo. a** The four GrpE patch mutations are mapped onto the structure (Gray and green, GrpE Mut-1 and Mut-4; Blue, GrpE Mut-2 and Mut-3). **b** Reactivation of the heat denatured luciferase (luc) by DnaK, DnaJ2, and WT and mutant GrpE. **c** ADP/ATP exchange of DnaK stimulated by WT and mutant GrpE. Control-1, no substrate peptide was added; Control-2, no ATP was added. **d** ITC isotherms from DnaK binding with either WT or mutant GrpE fitted to a single-site model. **e** Sequence alignment in the GrpE Mut-1 region from different organisms. Orange represents conserved residues, while other colors represent the mutation residues. **f** Strain (ΔgrpE attB::PdnaK-grpE kan) was subjected to marker exchange with streptomycin marked attB integrating plasmids and transformed with plasmids encoding empty vector, Mtb ALFA-GrpE WT, or Mtb ALFA-GrpE Mut-1. **g** Strains carry an ATc inducible CRISPRi construct targeting the endogenous GrpE, and an additional attB integrated CRISPRi resistant allele of the dnaK operon encoding: no Mtb GrpE, WT Mtb GrpE, WT Mtb GrpE-ALFA, Mtb GrpE-ALFA Mut-1, -2, -3, Mtb GrpE-ALFA Mut-1, Mtb GrpE-ALFA Mut-2, Mtb GrpE-ALFA Mut-3. 10-fold serial dilutions on agar containing no ATc (left panel) or ATc-50ng/mL (right panel). **h** Lysates probed for GrpE-ALFA (Nanobody_{ALFA}), RNAPβ, GrpE, and DnaK. **i** Immunoprecipitations of GrpE-ALFA using Nanobody_{ALFA} from lysates from panel (**g**) probed for GrpE and DnaK. **j** Quantitation of three replicates of DnaK co-immunoprecipitation with GrpE-ALFA WT and mutant proteins. The y axis is the ratio of DnaK signal in eluates to DnaK signal in the lysate starting material shown as percent of WT. **k** Tet-inducible Overexpression strains: Empty vector control, Ms GrpE, Mtb ALFA-GrpE_{wt}, Mtb ALFA-GrpE Mut-1, -2, -3, Mtb ALFA-GrpE Mut-1, Mtb ALFA-GrpE Mut-2, Mtb ALFA-GrpE Mut-3. Lysates probed for RNAPβ, GrpE-ALFA, DnaK, GrpE and mCitrine. *$P < 0.0001$ for the indicated comparisons. Data are presented as mean values +/− SD. $n = 3$ biologically independent samples. Ordinary one-way ANOVA was used for statistical analysis. Source data for panels (**b**–**d**) and (**h**–**k**) are provided as a Source data file.

consistent with the luciferase reactivation activity. We found the DnaK Mut-2 region is also well conserved (Fig. 5d), underscoring the functional importance of this region.

To assess the in vivo function of these DnaK residues, we applied the same approach as with GrpE mutated alleles. Marker exchange and CRISPRi confirmed the ability of wild-type Mtb ALFA tagged *dnaK* in the Mtb *dnaK* operon to support growth. For the mutant DnaK alleles, we found that although Mut-1 patch ALFA tagged *dnaK* supported growth, Mut-2, 3, or 4 did not (Fig. 5e, f, Supplementary Fig. 8b). To confirm in vivo expression of mutated Mtb DnaK proteins, we utilized the ALFA tag to differentiate Mtb DnaK from Ms DnaK. We found that all ALFA tagged Mtb DnaKs were expressed at similar levels (Fig. 5g), suggesting that protein levels were unlikely to explain the lack of essential function from DnaK Mut-2, Mut-3, or Mut-4. Although co-immunoprecipitation of GrpE by DnaK was detectable above control cells, there was no difference between DnaK mutant alleles and WT DnaK, indicating that even nonfunctional DnaK alleles are still able to physically interact with GrpE (Fig. 5h, i). We suggest that the lethal DnaK mutations (Mut-3 and Mut-4) that had no influence on GrpE binding may affect the allosteric coupling between DnaK NBD and SBD or affect DnaK interaction with DnaJ1/DnaJ2.

## The GrpE dimer opens the DnaK nucleotide-binding pocket

Superimposition of DnaK structures of the Mtb, *E. coli* and *G. kaustophilus* shows that the nucleotide-binding pocket is opened wider in the Mtb protein (Fig. 6a). We further compared the structures of the GrpE-bound Mtb DnaK with that of ADP-bound[37], ATP-bound[38] and nucleotide-free DnaK structures[39], and found that GrpE induces a 13° rotation of DnaK IB and a 6° rotation of DnaK IIB relative to the ADP-bound DnaK, and a 17° rotation of DnaK IB and a 5° rotation of DnaK IIB relative to the ATP-bound DnaK, while compared to the nucleotide-free DnaK, binding of GrpE induces a 10° rotation of DnaK IB and a 5° rotation of DnaK IIB (Fig. 6b). In the *E. coli* GrpE−DnaK−NBD complex, a 14° rotation of subdomain IIB with respect to the Hsc70 NBD were observed, which is consistent with our observations. In this conformation, the binding sites for the adenosine moiety on NBD IIB and the catalytic and triphosphate binding residues on lobe I of the DnaK NBD are wide apart, diminishing affinity for ADP and thus facilitating nucleotide exchange. This conformational switch resembles those found in the mammalian Hsc70 ATPase and their bacterial homolog DnaK ATPase upon binding to their NEFs (Bag in mammals and GrpE in bacteria), while in yeast Hsp70 ATPase, the contacts between the helical bundle domain of Hsp110 and subdomain IIB of Hsp70 stabilize the NBD in an "open nucleotide-binding cleft" conformation[23,24,40,41].

The topology of *Thermus thermophilus* GrpE was previously shown to be different from the *E. coli* GrpE[31]. However, we found here that the asymmetric Mtb GrpE dimer is topologically similar to the *E. coli* GrpE (Supplementary Fig. 10a). The helicity in the GrpE molecule A broke between Leu77 and Gln81, while the GrpE molecule B loses its

helicity between Lys88 and Ser90. Superposition of GrpE of the Mtb, *E. coli*, *G. kaustophilus*, and *T. thermophilus* reveals an overall similarity in the region of the proximal β-sheet but a greater curvature in the long N-terminal α-helix of Mtb GrpE (Fig. 6d, Supplementary Fig. 10a). The difference is likely a result of binding by the full-length DnaK in our structure. Further, the long α-helix that binds to DnaK NBD is bent more than the other long α-helix interacting with DnaK SBD. To accommodate the more bent shape of the Mtb GrpE, DnaK SBD shifts down by 10 Å compared to the *G. kaustophilus* structure (Fig. 6e).

## The DnaK−GrpE structure is dynamic and undergoes a series of defined motions

The modest resolution of the EM map (3.7 Å) suggested the presence of conformational heterogeneity in the DnaK−GrpE complex. We next conducted a multi-body refinement, an approach that probes structural heterogeneity and dynamic movement of a molecular complex[42]. We split the consensus map into four independent bodies: GrpE dimer, DnaK NBD lobe I, DnaK NBD lobe II, and SBD, and identified three types of movement characterized by three eigenvectors. The first eigenvector revealed a ratcheting motion of up to 20° of the GrpE dimer that is coupled with a 3 Å up-and-down movement of the DnaK SBD (Fig. 7a, Supplementary Movie 1). The second eigenvector shows a reciprocal rotation of the GrpE dimer (6°) and the DnaK NBD lobe II (12°) (Fig. 7a, Supplementary Movie 2). The third eigenvector represents a 25° rotation of the DnaK NBD lobe II (Fig. 7a, Supplementary Movie 3). Therefore, rather than being locked in a static open DnaK NBD conformation for nucleotide exchange, the Mtb DnaK−GrpE complex undergoes a series of defined motions that are centered around and coupled with the ratcheting of the GrpE dimer. The large-scale rotation of the DnaK NBD lobe II against lobe I clearly facilitates the ADP exchange for ATP. Thus, our use of full-length GrpE and DnaK reveals the coupling of the ratcheting of GrpE with the movement of the DnaK SBD. This observation raises the possibility that the function of the GrpE dimer may go beyond the established nucleotide exchange and may further include the release of the folded substrate bound to the DnaK SBD.

We next investigated the influence of nucleotide on the binding affinity of GrpE to DnaK by ITC (Fig. 7b). Consistent with our predictions, we found GrpE has similar binding affinity to the ADP-bound DnaK compared to the apo DnaK, while in the presence of ATP, the binding affinity is lowered by 10−15 times, indicating that GrpE binds to DnaK in the ADP-bound state, which causes the release of ADP and binding of ATP to restart the catalytic cycle.

Because the DnaK SBD binds to the N-terminal half of the GrpE long helix, we wondered if the preceding N-terminal 42-residue disordered peptide contributes to the GrpE-stimulated substrate release from DnaK. We made an N-terminal truncation (aa 43−235, ΔN42) GrpE protein and examined its effect on substrate release from DnaK by fluorescence polarization experiments. Interestingly, GrpEΔN42

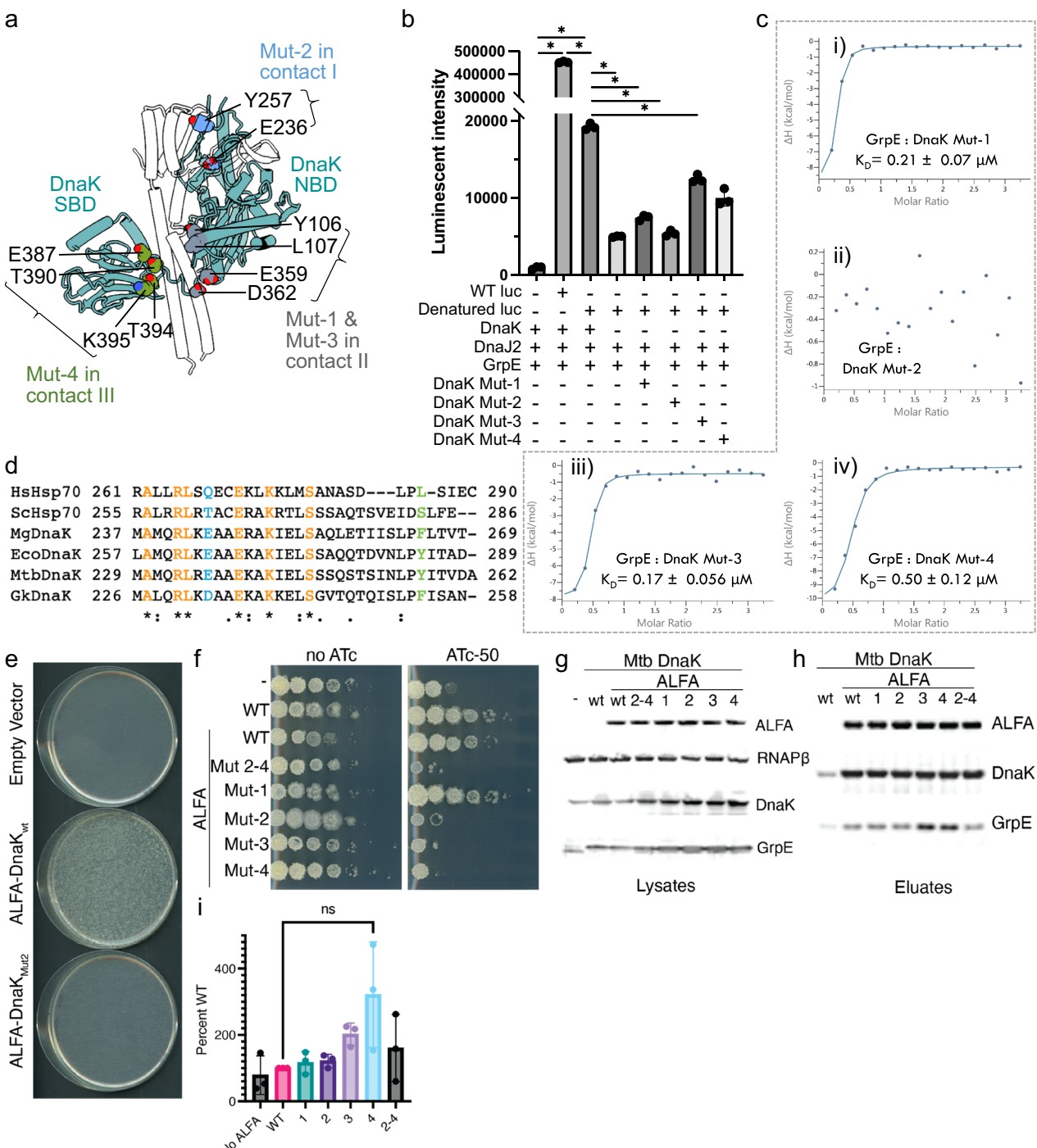

**Fig. 5 | DnaK mutations affect cell viability and interaction with GrpE both in vitro and in vivo. a** DnaK and GrpE are drawn as cartoon and are colored according to patch mutations (Gray, DnaK Mut-1 and -3; Blue, DnaK Mut-2; Green, DnaK Mut-4). **b** Reactivation of the heat denatured luciferase (luc) by DnaJ2, GrpE, and WT or mutant DnaK. **c** ITC isotherms from GrpE binding to the mutant DnaK fitted to a single-site model. **d** Sequence alignment in the DnaK Mut-2 region from different organisms. Orange represents conserved residues, while other colors represent the mutation residues. **e** *M. smegmatis* (Δ*dnaK attB::dnaK-twinstrep kan*) was subjected to marker exchange with streptomycin marked *attB* integrating plasmids encoding: empty vector, Mtb ALFA-DnaK WT, or Mtb ALFA-DnaK Mut-2. **f** Each strain carries an ATc inducible CRISPRi construct targeting the endogenous DnaK, and also carrying an additional *attB* integrated CRISPRi resistant allele of the *dnaK* operon as specified: no Mtb DnaK, WT Mtb DnaK, WT Mtb ALFA-DnaK, Mtb ALFA-DnaK Mut-2, -3, -4, Mtb ALFA-DnaK Mut-1, Mtb ALFA-DnaK Mut-2, Mtb

ALFA-DnaK Mut-3, Mtb ALFA-DnaK Mut-4. 10-fold serial dilutions on agar containing no ATc (left panel) or ATc-50ng/mL (right panel). **g** Lysates prepared from CRISPRi depleted strains probed for ALFA-DnaK (Nanobody$_{ALFA}$), RNAPβ, DnaK, and GrpE. **h** Immunoprecipitations of ALFA-DnaK using Nanobody$_{ALFA}$ coupled magnetic beads on lysates from panel (**f**) (note distinct lane order in **g** vs **h**). Eluates probed for ALFA-DnaK (Nanobody$_{ALFA}$), DnaK and GrpE. **i** Quantitation of three replicates of GrpE co-immunoprecipitation with ALFA-DnaK. Ratio of GrpE signal in eluates to GrpE signal in the lysate starting material shown as percent of WT ALFA-DnaK. GrpE co-immunoprecipitation did not differ significantly between samples (*P* values > 0.1). Data are presented as mean values +/− SD. *$P < 0.0001$ for the indicated comparisons. Ordinary one-way ANOVA was used for statistical analysis. $n = 3$ biologically independent samples. Source data for panels (**b**), (**c**) and (**g**–**i**) are provided as a Source data file.

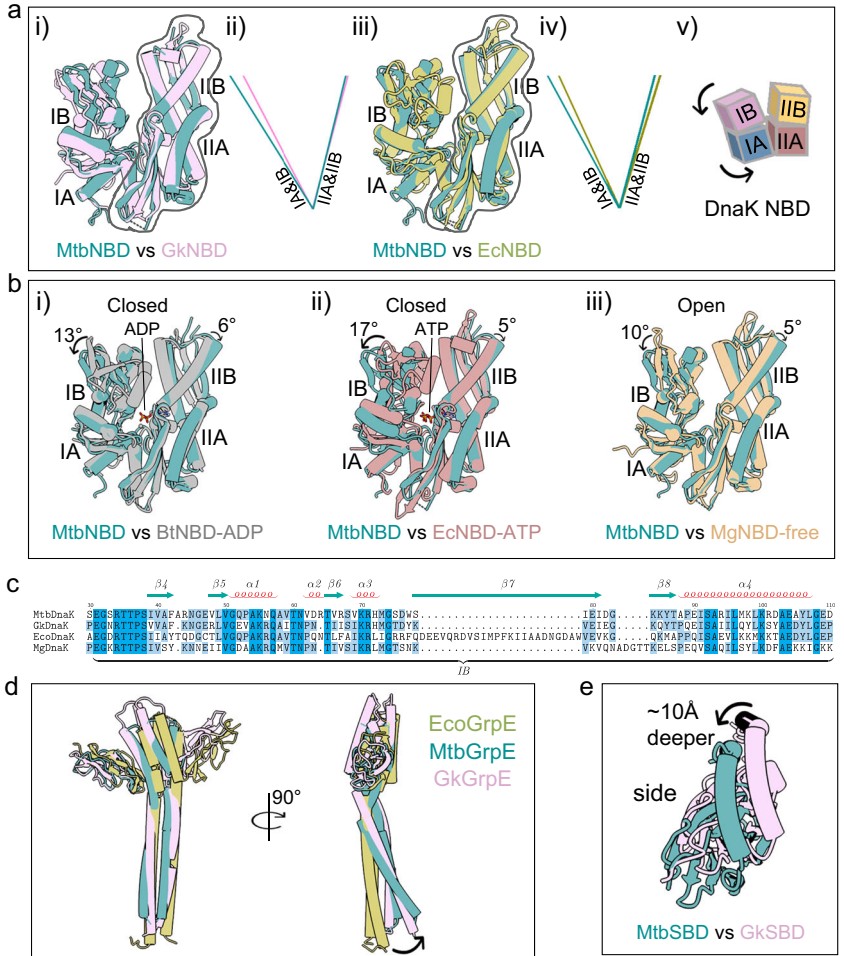

**Fig. 6 | Conformational changes in the DnaK NBD induced by the GrpE dimer.**
**a** (i) Superimposition of DnaK NBDs of Mtb and *G. kaustophilus* (PDB ID 4ANI). (ii) Comparison of the angles between lobes I and II of these two open nucleotide-binding pockets. (iii) Superimposition of DnaK NBDs of Mtb and *E. coli* (PDB ID 1DKG). (iv) Comparison of the angles between lobes I and II of the two open nucleotide-binding pockets in panel iii. The lines in panels ii and iv run through the length of compared lobes I and II. (v) Illustration of the subdomain movement with lobe II held constant. **b** Superimposition of DnaK NBDs of Mtb in open state and *Bos taurus* in the ADP-bound closed state (PDB ID 2QWL) (i), *E. coli* in the ATP-bound

closed state (PDB ID 7KO2) (ii), and *M. genitalium* in apo open state (PDB ID 5OBX) (iii). These structures are superimposed by their respective lobe II. The black curved arrows in all panels indicate subdomain movement. Rotation angles related to subdomain movement were labeled. **c** Sequence alignment of the DnaK IB domain of Mtb (accession number P9WMJ9), *G. kaustophilus* (accession number Q5KWZ7), *E. coli* (accession number P0A6Y8), and *M. genitalium* (accession number P47547). **d** Superimposition of GrpE of Mtb, *E. coli* (PDB ID 1DKG), and *G. kaustophilus* (PDB ID 4ANI). **e** Superimposition of DnaK-SBDs of Mtb and *G. kaustophilus* (PDB ID 4ANI). The black curved arrows in all panels indicate subdomain movement.

blocked substrate release from DnaK (Fig. 7c). In comparison, the GrpE Mut-1 to Mut-4 with intact N-tail competed with F-NR less effectively for binding to DnaK than the WT GrpE, but more effectively than the N-tail truncated GrpE (Supplementary Fig. 12). These results suggest that the disordered N-tail is critical for GrpE to allosterically regulate substrate release from DnaK.

## Discussion

In this study, we utilized cryo-EM and functional assays to understand how the Mtb GrpE helps DnaK to function. We discovered that the GrpE long N-terminal α-helix plays a dual role in the asymmetric binding between DnaK and GrpE that stabilizes the 1:2 stoichiometric complex and in a potential allosteric control of the DnaK NBD and SBD. Such GrpE-mediated allostery is above and beyond the well-established ATP- and substrate-binding induced DnaK allostery. We further found that the DnaK−GrpE complex is highly dynamic and undergoes several coordinated motions. We suggest that these motions underlie the function of the GrpE as a nucleotide exchanger and are responsible for the allosteric regulation of the DnaK.

## Asymmetry

Our biochemical and structural study of the Mtb system confirms the 1:2 stoichiometry of DnaK and GrpE first reported in the *E. coli* system but disputes the 2:2 stoichiometry from the crystallographic study of the *G. kaustophilus* system[23,24]. The asymmetric binding may provide a rationale for a single DnaK binding to the GrpE dimer. The DnaK NBD interacts with both the proximal β-sheet domain and the N-terminal long α-helix of one GrpE, while the DnaK SBD interacts with the N-terminal long α-helix of the other GrpE. In fact, the Arg64 and Arg75 in the GrpE N-terminal long α-helix contribute to interactions with both DnaK NBD and SBD. Their mutation (Mut-4) and GrpE patch Mut-1, which includes this double mutant, had the most severe effect on luciferase reactivation, ADP/ATP exchange rate, and binding affinity to DnaK, equivalent to omitting the DnaK in the refolding assay (Figs. 4–5). In agreement with these in vitro effects, GrpE Mut-1 was unable to support the essential DnaK function for bacterial viability and disrupt the DnaK interaction in vivo. Reciprocally, we found key DnaK residues interacting with the proximal β-sheet domain of GrpE are essential both in vitro and in vivo, as DnaK Mut-2 patch greatly reduced the luciferase reactivation activity and

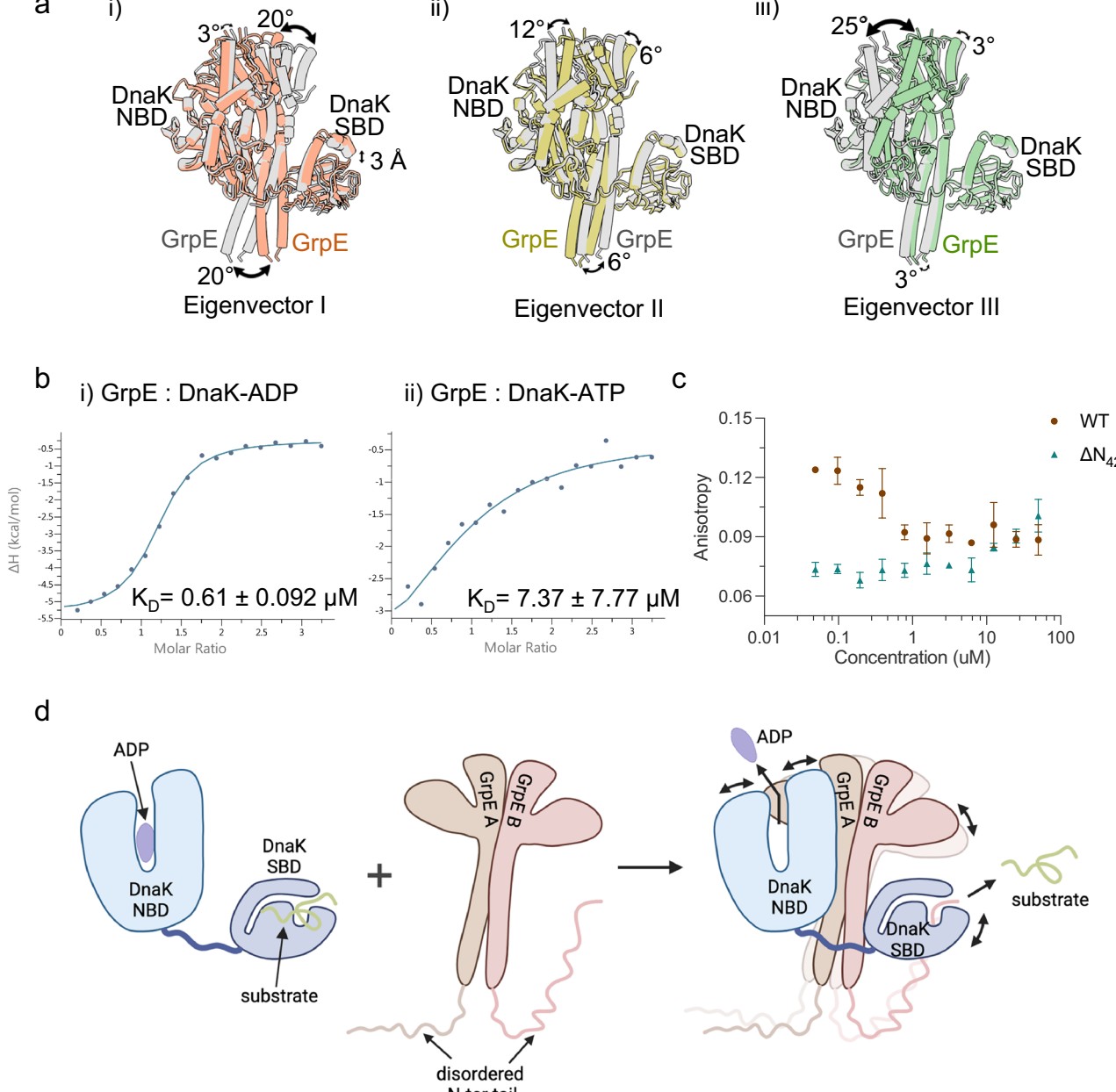

**Fig. 7 | Multi-body refinement reveals large domain movement in the Mtb DnaK–GrpE complex. a** Eigenvector I from multi-body analysis reveals a large-scale 20° rotation of the GrpE dimer (i), eigenvector II reveals a 12° opening of the DnaK NBD lobe I versus lobe II coupled with a 6° rotation of the GrpE dimer (ii), and eigenvector III reveals a large-scale 25° opening of the DnaK NBD lobe I versus lobe II coupled with a 3° rotation of the GrpE dimer (iii). Gray and color cartoons are before and after applying dynamic analysis. **b** ITC isotherms of Mtb DnaK binding to GrpE in the presence of ADP (i) or ATP (ii)

fitted to a single-site model. **c** Fluorescence polarization measurement of the F-NR binding to DnaK in the presence of an increasing concentration of WT and N-terminal truncated GrpE proteins (ΔN42). Source data are provided as a Source data file. **d** Sketch illustrating that the rich dynamics of the GrpE dimer, including the ratcheting motions of the dimer and the presence of the disordered N-terminal tail leads to a range of movements in DnaK NBD and SBD, stimulating ADP release from the NBD as well as substrate peptide release from the SBD. Created with BioRender.com.

blocked the in vitro interaction with GrpE and was unable to support growth in vivo.

## Dynamics

We observed in the consensus structure that the Mtb GrpE β-sheet domain wedges between the DnaK NBD subdomains IB and IIB, thereby widening the nucleotide-binding pocket to facilitate the nucleotide exchange. This nucleotide exchange mechanism resembles those observed in the homologous bacterial DnaK/GrpE systems, as well as in mammalian Hsc70/Bag and yeast Hsp70/Hsp110 systems in

which the three-helix bundle of the Bag or the β-sandwich domain of the Hsp110 wedges open the NBD of the Hsp70/Hsc70[23,24,40,41]. Therefore, despite the architectural difference, there appears to be a functional convergence in triggering the conformational switch in the Hsp70 proteins for nucleotide exchange. Interestingly, the coiled-coil middle domain of the ClpB also binds to the cleft between NBD subdomains IB and IIB of the DnaK in the Mtb and *T. thermophilus*[29,43,44], raising the probabilities that GrpE may compete with ClpB for stimulating the DnaK activity, or that ClpB may possess an unappreciated nucleotide exchange function for DnaK.

An important and unexpected finding of this study is the dynamics of the complex. Rather than capturing a static picture of the GrpE dimer stabilizing an open conformation of the DnaK NBD for nucleotide exchange, our study reveals that both the GrpE dimer and the DnaK are highly dynamic and that the motions of these two proteins are coordinated. The GrpE dimer appears to be constantly ratcheting. And the upper half of the ratcheting GrpE, the four-helix bundle and the β-sheet domain, induces a corresponding ratchet-like motion of the DnaK NBD lobe II, while the lower half of the GrpE dimer, the two parallel N-terminal helix, induces an up-and-down displacement of the DnaK SBD. The motion of the DnaK NDB lobe II on the top undulates the opening and closure of the nucleotide-binding pocket, and the displacement of the DnaK SBD at the bottom renders the domain partially flexible and may undulate the opening and closure substrate peptide pocket.

## Potential allostery

It has been widely appreciated that DnaK is allosterically regulated at two levels. At the first level, DnaK is allosterically regulated by ATP binding: In the compact configuration of DnaK, ATP binding in the NBD allosterically opens the polypeptide-binding site in the SBD via multiple contacts between the two domains[14,38]. At the second level, DnaJ allosterically regulates DnaK. As shown by the crystal structure of *E. coli* DnaJ-DnaK complex[22], the J-domain packs against the conserved interdomain linker and interacts with both DnaK NBD and SBD to upregulate ATP hydrolysis rate and downregulate nucleotide exchange rate[22].

Our study suggests another level of potential allosteric regulation of DnaK by GrpE. Upon ATP hydrolysis by NBD and substrate folding in SBD, the NBD and SBD are separated and connected only by a flexible linker. The GrpE dimer engages DnaK in this extended form, with the two long N-terminal α-helix providing a rigid connection and perhaps mediating allosteric communication between the DnaK NBD and SBD. Indeed, mutations that disrupt the interactions between the long N-terminal α-helix (GrpE Mut-1, Mut-4) and DnaK NBD (DnaK Mut-1, Mut-3) or DnaK SBD (DnaK Mut-4) all had adverse effects on the luciferase reactivation activity and substrate release efficiency in vitro. And key interface mutations (GrpE Mut-1) and DnaK NBD (DnaK Mut-3) or DnaK SBD (DnaK Mut-4) are lethal to the growth of *M. smegmatis*.

Interestingly, we found the GrpE disordered N-terminal tail is critical in stimulating substrate release from the Mtb DnaK, as removing this GrpE peptide blocked substrate release from DnaK. Based on the observed dynamics of the DnaK–GrpE complex and the critical role of the GrpE N-terminus, we propose the following working model for the Mtb GrpE-DnaK system (Fig. 7e). The GrpE dimer undergoes constant ratcheting motion to induce an opening of the DnaK NBD for ADP release and subsequent ATP binding. The GrpE ratcheting also enhances the dynamics of its disordered N-terminal peptide and induces an up-and-down displacement of the DnaK SBD. As a result of the enhanced dynamics, the GrpE N-terminus competes with the substrate peptide and facilitate substrate release from the DnaK SBD. In this model, the GrpE allosterically regulates DnaK, leading to a concomitant release of the folded substrate from the DnaK SBD and the ADP from the DnaK NBD. The potential GrpE-mediated DnaK allostery is consistent with our observation that the α-lid subdomain of the SBD is only partially resolved suggesting that the α-lid likely flexes between an open and closed state. Indeed, we found that telaprevir, a peptide substrate-binding inhibitor of the DnaK[30], failed to bind in the DnaK–GrpE complex, indicating the GrpE bound DnaK SBD is in a state that is incompatible with substrate binding.

To conclude, our study paints a dynamic picture of the Mtb GrpE-DnaK system and reveals how GrpE may facilitate nucleotide change and substrate release in DnaK. Given the high sequence and structure conservation, we expect that the dynamic mechanism observed in the Mtb system generalizes to the homologous systems.

## Methods

### Bacterial and DNA manipulations

Standard procedures were used to manipulate recombinant DNA and to transform *E. coli*. *M. smegmatis* strains were derivatives of *mc²155*[45]. All strains used in this study are listed in Supplementary Data 1. Plasmids including relevant features, and primers are listed in Supplementary Data 2 and 3. *M. smegmatis* was transformed by electroporation (2500 V, 2.5 µF, 1000 Ω). All *M. smegmatis* strains were cultured in LB with 0.5% glycerol, 0.5% dextrose (LBsmeg). 0.05% Tween$_{80}$ was added to all liquid media. For CRISPRi depletions anhydrotetracycline (ATc) was added at a concentration of 50 ng/mL. Antibiotic concentrations used for selection of *M. smegmatis* strains were as follows: kanamycin 20 µg/mL, hygromycin 50 µg/mL, streptomycin 20 µg/mL.

Plasmids for expressing full-length DnaK, DnaJ2, and GrpE were from a previous study[28]. To make alanine mutations in DnaK (The DnaK Mut-1 contains Y106A and L107A; Mut-2 contains E236A and Y257A; Mut-3 contains E359A and D362A; and Mut4 contains E387A, T390A, T394A, and K395A) and GrpE (GrpE Mut-1 in the aa 60–80 region contains R64A, N71A, R73A, K74A, R75A, and R78A; Mut-2 in the aa 110–120 region contains D110A and P115A; and Mut-3 in the aa 160–190 region contains S160A, R169A, and H181A; Mut-4 in the aa 60–80 region contains R64A and R75A) for in vitro activity assays, we purchased from Eurofins DNA oligonucleotides and custom synthesized gene fragments containing the targeted mutations, and PCR amplified the genes of the left regions with Phusion polymerase. PCR products and the DNA fragments containing the mutations were assembled using NEBuider HiFi DNA assembly master mix. All enzymes were purchased from New England Biolabs. Calcium competent *E. coli* Omnimax cells were transformed with the assembly products. The plasmids were sequenced by GENEWIZ to confirm the veracity of their cloned sequences. Confirmed plasmids were introduced into *E. coli* BL21(DE3) cells for protein production.

### Protein production and purification

For purification of WT DnaK and GrpE, *E. coli* BL21(DE3) strains transformed with the respective plasmids were grown in LB medium supplemented with 100 µg/mL carbenicillin at 37 °C until the OD$_{600}$ reached 0.6. Protein production was induced by isopropyl β-D-1-thiogalactopyranoside (IPTG) at a final concentration of 1 mM and continued cell growth for additional 4.5 h. Cells were collected by centrifugation and resuspended in buffer A (20 mM Tris, pH 8.0, 200 mM NaCl, 10% glycerol, 10 mM MgCl$_2$). We added a pinch of powdered DNase I & lysozyme and 1 mL saturated PMSF (Thermo-Fisher) to the suspension and lysed the cells by French press at 800 bars. The lysed cells were centrifuged at 36,600 × *g*. Supernatant was loaded onto a 5-mL Ni$^{2+}$-nitrilotriacetate acid (Ni-NTA) agarose column (GE Healthcare) equilibrated with buffer A. The column was washed with 30 mM imidazole and the target protein was eluted by buffer B (20 mM Tris, pH 8.0, 200 mM NaCl, 10% glycerol, 10 mM MgCl$_2$, 250 mM imidazole). Peak fractions containing DnaK were collected, pooled, and dialyzed against 1 L buffer A overnight while the SUMO-His-tag was cleaved by Ulp1. We then placed the sample in fresh dialysis buffer and dialyzed the sample for another hour. The SUMO-His-tag cleaved protein was passed through a 5-mL Ni-NTA prepacked column, concentrated, and applied to a HiLoad 16/60 Superdex 200 pg gel filtration column (GE Healthcare) equilibrated with buffer C (20 mM Tris, pH 8.0, 200 mM NaCl, 10 mM MgCl$_2$). Elution peaks were collected and concentrated to 10 mg/mL. Protein purity was examined by SDS-PAGE. DnaK mutants (Mut-1, Mut-2, Mut-3, and Mut-4) and GrpE mutants (Mut-1, Mut-2, Mut-3, Mut-4, and Mut-5) were purified similarly (Supplementary Fig. 11).

The Mtb DnaJ2 was purified as previously described[46]. Briefly, 6 L of *E. coli* BL21(DE3) cells were grown at 37 °C with agitation. When the OD$_{600}$ of the cell culture reached 0.6, 0.2 mM IPTG was added to

induce protein expression and the culture continued at 16 °C overnight. The cells were harvested by centrifugation and lysed with a French press. The lysate was centrifuged at 36,600 × g then the soluble supernatant was passed through a 5-mL Ni-NTA column. The bound protein was first washed with buffer A containing 50 mM imidazole, then eluted using buffer containing 250 mM imidazole. Peak fractions of Mtb DnaJ2 were collected and dialyzed against 2 L buffer A while SUMO-His-tag was cleaved with Ulp1. Cleaved proteins were passed through a 5-mL Ni-NTA prepacked column, concentrated, and loaded onto a HiLoad 16/60 Superdex 200 PG gel filtration column in buffer C. Protein fractions were collected and concentrated to 7.4 mg/mL.

## Cryo-EM

Purified DnaK (2 μM), DnaJ2 (4 μM), GrpE (4 μM) and a previously reported DnaK inhibitor telaprevir (200 μM)[30] were mixed in buffer D (25 mM Tris, pH 8.0, 40 mM NaCl, 10 mM MgCl$_2$). The mixture was incubated on ice for 40 min. 3 μL sample was then applied to a holey carbon grid (Quantifoil Cu R2/1, 300 mesh) that were freshly glow-discharged for 30 s. The grids were blotted with a piece of Whatman 595 filter paper with the blot force set to 3 s and blot time set to 3 s. The blotted grids were flash-frozen in liquid ethane using FEI Vitrobot Mark IV with the chamber temperature set to 10 °C and the humidity set to 100%. Cryo-EM images were collected in a FEI Titan Krios electron microscope (Thermo Fisher Scientific, operated at 300 kV) equipped with a Gatan K3 summit direct detector and a GIF quantum energy filter. Data acquisition was performed with SerialEM[47] in the super-resolution mode at a magnification of 105,000 and a pixel size of 0.414 Å per pixel. The objective lens defocus value was varied from −1.2 to −1.8 μm. During 1.5-s exposure, 60 frames (0.03 s per frame and the dose of 0.88 e/frame/Å$^2$) were collected with a total dose of 52.8 electrons Å$^{-2}$. To examine the possible nucleotide binding, we either added 2 mM ADP in the mixture of DnaK (2 μM), DnaJ2 (4 μM), GrpE (4 μM), DnaK inhibitor telaprevir (200 μM) or added 0.1 mM ADP and 0.1 mM ATP to the mixture of DnaK (4 μM) and GrpE (8 μM) and incubated the mixtures on ice for 40 min before making cryo-EM grids. Cryo-EM images were recorded on a Gatan K2 camera at a magnification corresponding to a pixel size of 0.58 Å in a Talos Arctica microscope (Thermo Fisher Scientific) operated at 200 kV. Thirty-frame movies were recorded with a dose rate of 0.2 electrons per Å$^2$ per second and an exposure time of 6 s.

## Image processing and 3D construction

We collected 15,720 raw movie micrographs for the DnaK−GrpE complex. All the micrographs were binned by a factor of two and drift corrected with electron-dose weighting in Motioncorr 2.1[48]. Contrast transfer function parameters of each aligned micrograph were estimated and the CTF effect was corrected using CTFFIND 4.1[49]. We used template picking to generate an initial dataset of 9,973,114 particles and performed 2D classification. Good 2D classes containing the DnaK−GrpE complex were selected, and a few selected averages were used to train Topaz[50], and the following Topaz particle picking yielded 628,553 particles. After 2D classification of this dataset, good 2D classes from template-based picking and the Topaz picking were combined, and duplicated particles were removed. A total of 289,063 particles were subjected for ab initio 3D reconstruction in cryoSPARC (version 3)[51], followed by heterogeneous refinement to generate five EM maps. Two best maps with intact DnaK−GrpE complex contained a total of 240,737 particle images were exported to the RELION format by UCSF PyEM [https://github.com/asarnow/pyem]. After 3D refinement, postprocessing, CTF refinement, and Bayesian polishing, the final EM map reached an overall resolution of 3.7 Å as estimated by the gold-standard Fourier shell correlation at a correlation cutoff value of 0.143 (Supplementary Fig. 2). The local resolution map was calculated using ResMap[52]. To improve the DnaK SBD density, we segmented the DnaK SBD density using Segger in UCSF Chimera[53], and used the

segmented DnaK-SBD map to generate a soft mask, and applied the mask for signal subtraction in RELION-3.0[54]. We also used PDB ID 4ANI to generate a low-pass filtered 3D map as a reference for 3D refinement of DnaK-SBD density. Both approaches resulted in similar and improved results. The separately refined DnaK SBD map was combined with the original map in ChimeraX to generate the final composite EM map. We used DeepEMhancer improve the visual presentation of the final map[55]. For the ADP-bound or ADP-ATP-bound samples, we collected 740 and 472 raw movie micrographs, respectively. All the micrographs were processed similarly. Template-based picking resulted in a small dataset of 598,315 particles for ADP-bound sample and a dataset of 388,559 particles for ADP/ATP-bound sample. 2D classification on the two datasets revealed a lack of complex formation between DnaK and GrpE in the presence of these nucleotides.

## Model building, refinement, and validation

We modeled the EM map of the Mtb DnaK−GrpE complex by referencing to the crystal structure of G. kaustophilus DnaK−GrpE complex (PDB ID 4ANI). We also used AlphaFold2[56] to predict the Mtb GrpE structure. The predicted GrpE model (aa 39–88 and aa 89–191 from chain A; aa 40–74 and aa 75–135 from chain B) were docked as individual rigid bodies into the EM map. The initial Mtb DnaK-NBD model was from PDB ID 4ANI, while the initial model of DnaK SBDβ was predicted by AlphaFold2. The DnaK linker region (aa 357–366 between NBD and SBD) was predicted by Robetta[57]. The atomic model of DnaK-NBD, DnaK-SBDβ, and linker were docked into the high-resolution EM map as rigid bodies, respectively. The DnaK SBDα was built in the density guided by the structure of G. kaustophilus DnaK (PDB ID 4ANI). All models were merged and subjected to manual adjustments in COOT[58], followed by real-space refinement in PHENIX[59]. The final model was validated using MolProbity[60,61]. Structural figures were generated in UCSF Chimera[53] and ChimeraX[62]. The contact areas between proteins were calculated from the atomic model in ChimeraX. Cryo-EM data collection, refinement, and validation statistics are listed in Supplementary Data 4.

## Multi-body refinement

For multi-body refinement, we used the particle images that had been motion corrected and radiation damage weighted in Relion[54]. We split the consensus refinement map into four separate bodies: the DnaK NBD lobe I, DnaK NBD lobe II, DnaK SBD, and the GrpE dimer. We relied on the atomic model to define the boundaries between the individual bodies and the separated and low pass filtered (10 Å) EM maps to generate masks to define the boundary with the solvent. The domain architecture of the complex implicates a possible rotation of the DnaK NBD lobe I and GrpE against each other, and another possible rotation of the DnaK NBD lobe II and DnaK SBD with respect to the GrpE dimer. We performed the Multi-body refinement using an initial angular sampling step of 1.8° and a translational sampling step of 0.25 pixel. We found that the structural variation can be partially accounted for by the first three eigenvectors (component 1 to 3), with components 1−3 accounting for 15.4%, 11.8%, and 10.2% of variance, respectively. Therefore, the first 3 eigenvectors explain 37.4% of the total variance in the dataset, and there is additional conformational heterogeneity that cannot be easily accounted for with the current method.

## Protein reactivation assay using denatured firefly luciferase

Protocol regarding this assay was modified from previous reports[28,29]. Briefly, denatured firefly luciferase (Fluc) was prepared by incubation of 500 nM native Fluc at 42 °C for 5 min in buffer E (50 mM Tris, pH 7.5, 150 mM KCl, 20 mM MgCl$_2$, and 2 mM DTT), followed by cooling down on ice for 5 min. Assisted folding reactions were performed by mixing chaperones and/or cofactors (6 μM DnaK or its mutants, 2 μM DnaJ2, and 2 μM GrpE or its mutants) together with denatured Fluc (final concentration 125 nM) in buffer D supplemented with 1 mg/mL BSA.

Reactions were initiated by adding 2 mM ATP to a final reaction volume of 100 µL and incubated at 25 °C for 30 min. Luminescence was measured by placing 10 µL reaction mixture into 90 µL of luciferase reagent (Promega) in a 96-well plate using a SpectraMax M2 luminometer. Native luciferase activity was measured using luciferase prepared in the same manner without the denaturation step. Reactions without DnaK were done in the same manner except for the addition of buffer instead of DnaK. A similar reaction without adding luciferase was conducted to measure the background signal. All reactions were done in triplicate. The luminescence readings were plotted using Prism 8 (GraphPad).

### ADP and ATP exchange assay

The GrpE-mediated ADP/ATP exchange activity of DnaK and the corresponding mutants were determined by measuring ADP and ATP levels using the ADP/ATP Ratio Assay Kit (Millipore Sigma). We used the substrate peptide NRLLLTG (NR) instead of firefly luciferase in this assay. We first mixed DnaK (6 µM), DnaJ2 (2 µM), and GrpE or its mutants (2 µM) with or without substrate peptide (60 µM) in buffer E supplemented with 1 mg/mL BSA. Reactions were initiated by adding 600 µM ATP to a final reaction volume of 100 µL and the tubes were incubated at 25 °C for 30 min. 10 µL reaction mixture was added to 90 µL ATP reagent in a 96-well plate and incubated for 1 min. Luminescence was measured on the SpectraMax M2 luminometer ($RLU_A$). After 10 min incubation, the luminescence for ATP ($RLU_B$) was read, followed by addition of 5 µL ADP reagent and 1 min incubation. Then Luminescence for ADP ($RLU_C$) was read. The ADP/ATP ratio was calculated as $(RLU_C - RLU_B)/RLU_A$. All reactions were done in triplicate.

### Isothermal titration calorimetry

Interactions of wild type and mutant DnaK with wild type and mutant GrpE were measured at 25 °C in a MicroCal PEAQ ITC device (Malvern). Concentrated DnaK, GrpE and their corresponding mutant proteins were first dialyzed against buffer F (20 mM Tris pH 8.0, 150 mM NaCl, and 1 mM β-mercaptoethanol) overnight in Slide-A-Lyzer 10 K dialysis cassettes. Wild type or mutants DnaK at 12 µM in a calorimeter cell was titrated by 19 consecutive injections (2 µL each with a 150 s waiting time) of 200 µM wild type or mutant GrpE. For testing the binding of DnaK with GrpE in the presence of ADP or ATP, everything is the same except for 2 mM ADP or 2 mM ATP were added to both DnaK and GrpE protein solutions. The binding isotherms were plotted using MicroCal PEAQ ITC analysis software. All reactions were done in duplicate. Averages and standard deviation were reported in Supplementary Table 1.

### Fluorescence anisotropy assay

NR is a model peptide substrate (sequence: NRLLLTG) of DnaK. We purchased a N-terminal fluorescein-labeled NR (F-NR) with a purity of >95% from PEPTIDE 2.0. 20 µM DnaK were incubated with 200 µM ADP and 50 nM F-NR in buffer E at room temperature for 30 min. Serial dilutions of GrpE WT or mutant protein (concentration ranging from 100 µM to 0.01 µM) were made in buffer E. To start the substrate release reaction, 15 µL GrpE serial dilution solution was added to 15 µL DnaK reaction mixture. Fluorescence polarization measurements were carried out on a BioTek Synergy Neo2 hybrid multimode reader. The readout data were fit to a nonlinear regression equation ([Inhibitor] vs. response with three parameters) using PRISM 10 (GraphPad) to obtain the $IC_{50}$ value. The experiments were repeated three times.

### Mass spectrometry

For sample preparation, 10 µM none-nucleotide (S01-1 to -4) or ADP (S02-1 to -4) or ATP (S03-1 to -4)-bound DnaK was mixed with 10 µM DnaJ2 that was pre-incubated with peptide (NRLLLTG) in buffer E and incubated for 30 min. 10 µM GrpE was added and further incubated for 10 min. All the unbound peptide and nucleotide were removed by gel filtration purification. Peak fractions corresponding to DnaK–GrpE complex were collected and concentrated. 90 µL of each sample was subjected to a metabolite extraction by the addition of 800 µL of ice-cold extraction solvent (40% acetonitrile, 40% methanol, 20% water v/v), pulse vortexed, sonicated in a water bath sonicator (5 min), and incubated on wet ice (60 min). Then, an additional 480 µL of water was added to aid nucleotide recovery. Finally, 1 mL of extract was dried in a vacuum evaporator and resuspended in 40 µL of water and 2 µL injected for LC-MS analysis. Full LC-MS parameters are available in Supplementary Data 5. Briefly, liquid chromatography was conducted on an Agilent 1290 Infinity II UHPLC using a Zorbax Extend C18 2.1 × 1500 mm 1.8-micron (#759700-902, Agilent Technologies). Mobile phases were A: 3% methanol in water and B: 97% methanol in water. Both mobile phases contained 10 mM tributylamine, 15 mM acetic acid, and 0.1% (v/v) medronic acid (#5191-4506, Agilent). Analytes were detected on a triple quadrupole mass spectrometer (Agilent 6470) in ESI negative mode using standard-verified transitions for NRLLLTG peptide (784.0 → 303.0, RT = 0.98 min), adenine (134.0 → 107.0, RT = 2.26 min), adenosine (266.0 → 107, RT = 4.60 min), adenosine monophosphate (346.0 → 79.0, RT = 6.76 min), adenosine diphosphate (426.0 → 328.0, RT = 7.73 min), and adenosine triphosphate (506.0 → 159.0, RT = 8.38 min). Positive controls used extract from a sample containing 10 µM DnaK-GrpE-DnaJ2, 100 µM each of NR, ADP, AMP, and ATP without running gel filtration. The gel filtration buffer (25 mM HEPES pH 7.5, 150 mM KCl, 2 mM DTT) was used as negative controls, as it contained neither nucleotide nor peptide substrate. Each experiment was repeated four times ($n = 4$) except positive and buffer controls which were repeated two times ($n = 2$). A total of 16 samples were analyzed and the results are listed in Supplementary Fig. 4.

### SEC-MALS

The gel filtration peak fractions for the DnaK–GrpE complex were collected, concentrated and loaded onto a Superose 6 increase 10/300 GL size exclusion column (GE Healthcare) in line with both a DAWN HELIOS II MALS detector (Wyatt Technology) and an Optilab T-rEX differential refractometer (Wyatt Technology). Differential refractive index and light scattering data were measured, and the data analyzed using ASTRA 6 software (Wyatt Technology). Extrapolation from Zimm plots was used to calculate molecular weights using a dn/dc value of 0.185 mL/g. Data were plotted in PRISM 10 (GraphPad).

### Nanobody purification and NHS-magnetic bead coupling

Nanobody ALFA[63] was cloned into a PET21b vector containing a PelB leader sequence and purified using 1 L BL21(DE3) in Terrific Broth supplemented with 100 µg/mL carbenicillin and induced overnight with 0.1 mM IPTG. Cells were harvested and nanobodies were purified from the periplasm as previously described[64]. Briefly, the periplasmic fraction was isolated by osmotic shock and bound to 2 mL nickel affinity resin (HisPur, Thermo Fisher Scientific). Resin was washed 5 times with 20 mL 20 mM Tris, pH 8, 150 mM NaCl, 10 mM Imidazole. Nanobodies were eluted 5 times with 1 mL 20 mM Tris, pH 8, 150 mM NaCl, 250 mM Imidazole. Pooled elution was dialyzed overnight in 3.5 K Slide-a-lyzer (Thermo Scientific) in PBS. Nanobodies were then concentrated using a 3.5 K Amicon Ultra (Millipore Sigma). Nanobodies were aliquoted and stored at a final concentration of 10 mg/mL in PBS with 10% glycerol at −80 °C. For magnetic bead coupling, 300 µL NHS-magnetic beads (Pierce) washed with 1 mL PBS were resuspended in 1 mL of 1 mg/mL Nanobody ALFA in PBS, 1% glycerol then vortexed for 30 s. The tube was then left mixing end over end for 18 h at 4 °C. Beads were collected and resuspended in 1 mL 1 M Tris pH 8 and incubated for 30 min at RT. Beads were collected and washed with 1 mL 50 mM tris, 150 mM NaCl, 10% glycerol, 4 mM EDTA. Beads were then washed 2 times with 1 mL PBS, 10% glycerol, 4 mM EDTA. Beads were resuspended in a final volume of 300 µL PBS, 10% glycerol, 4 mM EDTA and stored at 4 °C prior to use.

## Immunoblotting

For protein and epitope tag detection, the following primary antibodies were used: ALFA (purified nanobody ALFA, 10 mg/mL 1:5000), DnaK (rabbit antisera, 1:10,000[65]), GrpE (rabbit antisera, 1:10,000[65]), mCitrine (Invitrogen, Mouse Anti-GFP monoclonal antibody, 1 mg/mL, 1:10,000), and RNAP-β (BioLegend, 8RB13 Mouse Anti-E. coli RNAPβ monoclonal, 1:10,000, Cat # 663905). Secondary antibodies used: anti-VHH-HRP (GenScript, 1:10,000, Cat# A01861-200), anti-mouse-HRP (Invitrogen, 1:5000, Cat # 62-6520), anti-rabbit-HRP (Invitrogen, 1:5000, Cat # 65-6120). Blots were imaged using SuperSignal West Pico Plus Chemiluminescent Substrate (Thermo Fisher Scientific) and an iBright FL1000.

## ALFA tag immunoprecipitations

Depletion strains were diluted in 15 mL LBsmeg supplemented with hygromycin and ATc-50 and grown for 18 h to a target $OD_{600}$ of 0.4. Cultures were cooled on ice then harvested by centrifugation $3700 \times g$ 10 min at 4 °C. All subsequent steps prior to elution were performed at 4 °C. Supernatant was discarded and cell pellets were resuspended in 1 mL cold PBS, 10% glycerol, 4 mM EDTA and transferred to 2 mL tubes. Cells were pelleted by centrifugation $10,000 \times g$ 1 min, and supernatant was discarded. Cells were resuspended in 500 μL PBS, 10% glycerol, 4 mM EDTA and lysed via bead beating (Mini-beadbeater-16; Biospec) 3 times for 45 seconds with 5 min on ice between. Beads, unbroken cells, and debris were pelleted at $10,000 \times g$ for 1 min. Supernatant was transferred to a new 1.7 mL microcentrifuge tube. The pellet was resuspended in 500 μL PBS, 10% glycerol, 4 mM EDTA and disrupted (Mini-beadbeater-16; Biospec) 3 times for 45 seconds with 5 min on ice between. Beads, unbroken cells, and debris were pelleted at $10,000 \times g$ for 1 min. Supernatant was pooled in the 1.7 mL microcentrifuge tube. This 1 mL lysate was pelleted at $10,000 \times g$ for 2 min. Supernatant was transferred to a new 1.7 mL tube and a 50 μL aliquot of lysate was taken as the starting material. To the remaining lysate 10 μL of NBALFA magnetic beads were added and tubes were left mixing on a wheel at 4 °C for 2 h. Beads were collected via a magnetic stand and washed 3 times with 1 mL PBS, 10% glycerol, 4 mM EDTA. Beads were suspended in 100 μL 2x sample loading buffer with 50 mM DTT and heated to 75 °C for 10 min for elution.

## GrpE overexpression

Expression strains were diluted in 15 mL LBsmeg supplemented with hygromycin and grown for 18 h to a target $OD_{600}$ of 0.2. Cultures were split into 2–15 mL volumes, and to one of the pair ATc-50 was added. Cultures were left to induce at 37 °C for 6 h. Cultures were cooled on ice then harvested by centrifugation $3700 \times g$ 10 min at 4 °C. All subsequent steps prior to sample buffer addition were performed at 4 °C. Supernatant was discarded and cells were resuspended in 1 mL PBS, 10% glycerol, 4 mM EDTA and transferred to a 2 mL O-ring tube. Cells were harvested by centrifugation at $10,000 \times g$ 1 min, and supernatant was discarded. Cells were resuspended in 250 μL PBS, 10% glycerol, 4 mM EDTA and lysed via bead beating (Mini-beadbeater-16; Biospec) 3 times for 45 s with 5 min on ice between. 250 μL of 2x sample buffer with 50 mM DTT was added and samples were heated to 85 °C for 5 min. Beads, unbroken cells, and debris were pelleted at $10,000 \times g$ for 1 min prior to loading.

**Statistics and reproducibility.** No statistical method was used to predetermine sample size. The data point corresponding to the first test run was excluded for the ITC experiments. The experiments were not randomized. The Investigators were not blinded to allocation during experiments and outcome assessment. PRISM 10 (GraphPad) was used to fit the luciferase assay, ADP/ATP exchange data, ITC data, and the fluorescence anisotropy data.

## Reporting summary

Further information on research design is available in the Nature Portfolio Reporting Summary linked to this article.

## Data availability

Cryo-EM 3D composite map of *M. tuberculosis* DnaK–GrpE complex at 3.7 Å average resolution has been deposited in the Electron Microscopy Data Bank under accession code EMDB-29912. The original map was deposited under accession code EMDB-29913. Focused map of DnaK SBD domain was deposited under accession code EMDB-29914. The corresponding atomic model has been deposited in the Protein Data Bank under accession code 8GB3. This study also used the crystal structure of the *E. coli* GrpE dimer bound to the DnaK NBD (PDB ID 1DKG), the published DnaK NBD structure of *G. kaustophilus* (PDB ID 4ANI), the DnaK NBD structure of *Bos taurus* in the ADP-bound closed state (PDB ID 2QWL), the *E. coli* DnaK NBD structure in the ATP-bound closed state (PDB ID 7KO2), the *M. genitalium* DnaK NBD in apo open state (PDB ID 5OBX). For sequence alignment, this study used the DnaK IB domain of Mtb (accession number P9WMJ9), *G. kaustophilus* (accession number Q5KWZ7), *E. coli* (accession number P0A6Y8), and *M. genitalium* (accession number P47547). Source data are provided with this paper.

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

## Acknowledgements
Cryo-EM data were collected at the David Van Andel Advanced Cryo-Electron Microscopy Suite at Van Andel Institute (VAI). We thank Gongpu Zhao and Xing Meng for help with EM data collection, Ryan Sheldon and Christine Isaguirre at the VAI Mass Spectrometry Core for help with mass spec analysis, Jayakrishnan Nandakumar and Jonathan Williams at University of Michigan for help with SEC-MALS experiment, and Xiang Feng and Hua Li at VAI for help with EM data processing. This study was supported by National Institutes of Health grants AI070285 (to H.L.) and AI138446 (M.S.G.).

## Author contributions
X.S.X., M.S.G., and H.L. conceived and designed experiments. X.S.X. and A.K. purified proteins. X.S.X. performed cryo-EM, image processing, 3D reconstruction, atomic model building. X.S.X. and A.K. performed in vitro activity assays. A.F. and P.S.M. performed and analyzed the *M. smegmatis* experiments. X.S.X., M.S.G., and H.L. analyzed the data and wrote the manuscript with input from all authors.

## Competing interests
M.S.G. has received consulting fees from Vedanta Biosciences, and Fimbrion Therapeutics and has equity in Vedanta biosciences. The rest of the authors have no competing interests.
