## [Peer Review File · Nature Communications]

Structure of the M. tuberculosis DnaKGrpE complex reveals
how key DnaK roles are controlledReviewer #1 (Remarks to the Author):

DnaK is the eubacterial version of the molecular chaperone Hsp70. The protein consists of a nucleotide binding domain (NBD) and a substrate binding domain (SBD), whose nucleotide state and binding properties are allosterically coupled. DnaK binds to folding intermediates and misfolded protein species, and supports their folding in an ATP hydrolysis driven conformational cycle, which is regulated by the cochaperones DnaJ and GrpE. DnaJ recruits DnaK-ATP to substrates and triggers ATP hydrolysis, which leads to tight substrate binding. GrpE binding to DnaK triggers release of ADP, allowing binding of ATP and substrate release. The previous crystal structure of the *E. coli* GrpE-NBD complex was asymmetric and showed how one subunit in the GrpE dimer wedged into the nucleotide binding cleft. The coiled-coil domain in GrpE contributed to NBD binding and a preceding flexible part to substrate release. In contrast to the observed 2:1 stoichiometry, a later crystal structure of a *Geobacillus kaustophilus* GrpE-DnaK complex suggested a 2:2 stoichiometry, or 4:4 if DnaK-DnaK pseudo-substrate interactions in the crystal lattice are considered.

The present manuscript now reports the cryoEM structure of a complex of DnaK and GrpE from *M. tuberculosis* at 3.7 Å resolution. The complex has 2:1 stoichiometry. The interaction with the NBD seems to resemble the interaction observed in the *E. coli* complex (Here, a structure comparison by superposition and the r.m.s.d. value would be informative.). In addition, an interaction of the SBD with the coiled-coil helix pair of GrpE is observed, apparently similar to the interaction with one of the DnaK copies in the structure of a *Geobacillus kaustophilus* GrpE-DnaK complex (Again a structure comparison like above would be informative.). The C-terminal part of the SBD seems poorly ordered/defined in the density. Whether a substrate or a substrate-mimicking inhibitor is bound or not is unclear. Presence or absence of this ligand in the complex with DnaK should be demonstrated independently.

Next, the authors use site-directed mutagenesis to probe the observed contact interfaces. Surprisingly, mutation of the contacts in the helix pair region in GrpE seemed to affect the affinity and nucleotide exchange activity more than mutations in the wedge and the 4-helix bundle, closer to the nucleotide binding cleft. In vivo, the former mutation did not support growth, in contrast to the latter. So, the contacts to the NBD and SBD close to the DnaK interdomain linker seem to contribute substantially to the GrpE binding affinity.

When mutations were introduced in the interface regions in DnaK, mutations in subdomain IIB (next to the wedge) had the strongest effect on affinity. This mutation disrupted DnaK function in vivo. Mutation of residues contacting the helix pair had GrpE affinity similar to wildtype, but were non-functional in vivo. Here it is not clear whether these mutations disrupt DnaK function by other means than interactions with GrpE, for example by disrupting the allosteric coupling between the DnaK domains or the interaction with DnaJ.

Finally, the authors study the dynamics of the complex by eigenvector analysis. The authors propose that the wobbling of the domains in the complex facilitates nucleotide and substrate release. This seems plausible to me. However, I would not call this "allostery". That the DnaK nucleotide binding cleft engages in alternative contacts in the complex with GrpE, which weaken nucleotide binding, I would describe as allostery. But the alpha-helical lid over the substrate binding pocket is modelled in place for substrate interactions unlike in the DnaK-ATP complex. If MtGrpE binding is substantially weakening the interaction of MtDnaK with substrate, an alternative mechanism seems to be required, for example displacement by a substrate-mimicking peptide in the disordered N-terminal tail. This could be investigated by an N-terminal truncation mutant of MtGrpE. To show that substrate release from DnaK is triggered by GrpE binding or that its affinity is lowered in the complex, a biochemical assay should be set up.

I am not sure if the present structure goes much beyond the previously existing structures. The more interesting parts - the DnaK SBD and its helical bundle and the beta-domain in the second GrpE copy - are rather ill-defined in the present density. The mutational analysis seems to be contradictory in parts. Experimental evidence for substrate release from DnaK via GrpE binding in *M. tuberculosis* is missing.

Detailed points:

How much of the N-terminal regions of GrpE could not be modelled in the complex? In case large portions are missing in the model, this should be mentioned in the main text. According to the PDB validation report also large C-terminal portions were not modelled in one of the GrpE subunits.

p. 3, line 86. What happens when DnaK-DnaJ2-GrpE are mixed in 2:2:2 ratio? Is only one copy of DnaK incorporated into the complex?

p. 4, line 119. Normally the lobes in DnaK are designated I and II and the respective subdomains A and B (and not vice versa).

Fig. 3 There are several contacts shown which are rather unlikely at neutral pH, for example hydrogen bonds between D362 and E57, or between R396 and R75. Can you please show how well the respective sidechains are defined in density? If they are not defined well, other side chain conformations seem more likely. Is the sequence register in the GrpE coiled-coil helices unambiguously clear?

Also, for some of the hydrogen bonds shown, the geometries seem unfavorable (for example from T60). The hydroxyl group of Y257 seems to engage in H-bonds to three carbonyl groups, exceeding its hydrogen bonding capability.

p.5, line 130 The references to the "lower left side" and "lower right side" do not work without a specific Figure for illustration. Better use a description independent of a specific view: The disjointed SBD domains exhibit diverse orientations with respect to the NBD in the available structures?

Fig. 4B The luciferase folding yield (~12%) appears very poor. Either the Mtb chaperones do not work well with this substrate, or the concentrations (or concentration ratios) of the proteins in the assay are poorly chosen. In the latter case, a clearer result for GrpE Mut-2 and Mut-3 might be obtained by concentration optimization.

Fig. 4D, panel ii) What was the approximate KD? The curve does appear interpretable.

Fig. 6A There are multiple structures of the E. coli DnaK in the PDB. It is not clear which one was chosen in subpanel i – please specify in the Figure legend.

In the GrpE-DnaK-NBD complex, the conformational change was described as a 14° rotation of subdomain IIB with respect to the Hsc70 NBD (ADP complex, I presume) in the first structure paper (Harrison et al., 1997). So, is there an additional outward rotation of subdomain IB in the M. tuberculosis complex? Please describe this more clearly in the main text.

The arrows in subpanel ii seem to point into the wrong direction.

Fig. 6B This superposition is pretty messy. The conformations cannot be distinguished easily. Better compare one by one. Was there a single subdomain in all structures held in place? To which structure pair does the curved arrow in subpanel i refer?

In the ATP- and ADP-bound states, the nucleotide binding pocket should be closed. Why is there an opening angle?

In the ATP state, the NBD lobes of Hsp70 are known to exhibit sideways shearing compared to the ADP state. So, there are rather complex conformational changes between the states, which cannot be adequately described with one angle.

For readers not familiar with Hsp70, it would be useful to include the nucleotide to show the position of the nucleotide binding pocket.

Reviewer #2 (Remarks to the Author):

The manuscript by xiao, et al. describes their work using cryo-EM and functional assays to understand how the Mtb GrpE helps DnaK to function in ADP and refolded substrate release. They found that the GrpE long N-terminal α -helix plays a dual role in the asymmetric binding to DnaK and the allosteric control of DnaK's NBD and SBD domains. They further found that the DnaK-GrpE complex is highly dynamic and undergoes coordinated motions that are unique both scientifically and technically. Although there is prior work on the GrpE complex with DnaK NBD, this work provides complete and dynamic views of how GrpE regulates DnaK's structure and function. The study represents an excellent example of showing the utility of cryo-EM to explore the structure and dynamics of protein complexes. Functional data support their structural analysis. The manuscript is well written although there are typos and confusing statements. Below are my comments for the authors to consider.

Scientific: The allosteric regulation of NBD and SBD by ATP binding and substrate release has been well-studied both structurally and functionally. The regulatory role of GrpE in accelerating ADP release after ATP hydrolysis in DnaK NBD is also well known. Using the full-length DnaK, the authors determined the GrpE-DnaK complex structure and propose that the GrpE dimer allosterically regulates the ADP release and substrate refolding. The solved complex structure does not contain ADP or refolded substrate. So, it is not clear to the reviewer if GrpE-mediated ADP release may regulate refolded substrate release and vice versa. Is it possible that the release of the refolded substrate may be triggered by ATP hydrolysis, but not ADP release, and thus prior to the GrpE binding? The authors should discuss other possible scenarios in the discussion session.

Technical: Cryo-EM is the ideal tool to study the dynamic structures of protein complexes. The GrpE-DnaK has a molecular mass of only about 100 kDa, which is relatively small for cryo-EM analysis. I have several technical comments concerning the cryo-EM analysis of this small complex.

1) The complex structure was reconstructed at 3.7 Å resolution. However, in the manuscript, there is a lack of validation of the reconstructed map. It's suggested that the authors provide FSC curves for masked and unmasked maps as well as a local-resolution map to show the quality of their reconstruction. In Figure 3, the authors show residue-residue interactions between GrpE and DnaK for the three contacts. However, there are no cryo-EM densities to support any of these interactions. The authors should provide cryo-EM densities or explanations on how they validate all or some of the interactions.

2) The Mtb DnaK SBD contains only 256 residues (from Supplementary Figure 4). The authors used the focused refinement to get a reconstruction of the DnaK-SBD at 5.8 Å resolution. The authors should validate their cryo-EM map by using FSC curves. In addition, they need to show the map quality and the fitness of the atomic model they built. The map quality for the SBD also relates to the quality of the model they used to draw GrpE-SBD interactions in Figure 3E. In addition, with a domain of 256aa, its projected signal is very weak. I wonder how strong the signal they can get after partial signal subtraction of other components. The authors should compare the maps before and after focused refinement to see if any improvement in the map quality.

3) The authors used multibody refinement and principal component analysis to derive motions in the GrpE-DnaK system. The total molecular mass of the system is about 100 kDa. The authors split the system into four bodies. The same to my concern above, the authors should validate their refined maps to see if they obtained better maps from the multi-body refinement. It's a question to me if Relion can perform reliable four-body refinements with individual bodies less than 100 kDa. The noise associated with partial signal subtraction makes the multibody refinement very challenging. Instead, have authors tried 3D variability analysis in cryoSPARC?

Specific comments:

1) The authors repeatedly use DnaK/Hsp70 in the introduction. They can mention Hsp70 once and use DnaK in the rest of the introduction.

2) Line 51: What is NEF? Please spell out the full name.

- 3) Lines 73-74: "This property contrasts with DnaK of the other bacteria that is nonessential." Does this mean that DnaK is dispensable in other bacterial species including *M. smegmatis* that was used in this study? Please cite a reference to support this statement.
- 4) Lines 139-142: Are the contact region III interface between GrpE B and DnaK SBD also charge-charge interactions? How was the contact area of the region III interface (891 Å²) calculated, from the atomic model or cryo-EM map?
- 5) Lines 157-164: What is the experimental evidence to support the interactions shown in Figure 3? Without showing cryo-EM densities, it's hard to evaluate the quality of the specific interactions described in the section from lines 157 to 164.
- 6) Line 178: did the authors add Mg²⁺ which is required for the ATP hydrolysis activity? If they did, they should include Mg²⁺ and its concentration here.
- 7) Line 210: "Furthermore, neither *Mtb* GrpE Mut-2 nor Mut-3 allele supported viability, demonstrating that these mutant proteins could complement for essential...". This sentence is confusing and contradicts the next sentence "However, GrpE Mut-1 (Fig. 4F) or GrpE Mut1-3 (data not shown) did not support viability, indicating that the Mut-1 residues are essential for DnaK system.". It reads that Mut-2 and Mut-3 are lethal. Please revise the sentence for consistency.
- 8) Line 212: "(data not shown)". I think the journal policy requires showing the data which could be presented as part of the supplementary data.
- 9) Line 216: It seems that DnaK is essential for *M. smegmatis* viability. This statement is confusing with the authors' argument of DnaK as a drug target because it's vital only to *Mtb*. The authors should revise their text to address such discrepancies.
- 10) Line 260: If Mut-1 is insoluble in vitro (Line 246), the authors should explain why it supported bacteria growth in vivo.
- 11) Lines 297-298: There are many factors contributing to a low or modest resolution of a cryo-EM map. What about air-water interface issue? The authors should include a local-resolution map and use it to suggest the presence of conformational heterogeneity.
- 12) Lines 333-334: The authors claim that Arg64 and Arg75 contribute to the interactions with both DnaK NBD and SBD. Their GrpE Mut-1 contains six mutations (R64A, N71A, R73A, K74A, R75A and R78A). Did they try to make an R64A-R75A double mutant to evaluate their importance in ADP release and/or substrate release?
- 13) Supplementary Figure 2. Missing a scale bar in the raw micrograph. "right corner" should be "left color". Eulerian angel distribution" should read as "Eulerian angular distribution".
- 14) Supplementary Figures 3& 4: Missing accession numbers for the GrpE and DanK homologs used in their multiple sequence alignments.

Reviewer #3 (Remarks to the Author):

In this study by Xiao et al, the authors use FL DnaK and GrpE from *M. Tub.* to understand the mechanism of binding/allostery that promotes ADP release from DnaK and thus facilitating nucleotide exchange required for rounds of substrate refolding by DnaK. The authors reconstitute the complex from recombinant components of DnaK:GrpE and use cryo-EM to determine a structure uncovering a 1:2 stoichiometry. The cryo-EM data is of reasonable quality but require additional processing and masking to interpret the dynamics and given that the resolution of the maps is not fantastic the conclusions drawn about the dynamics (and small changes in angular deviation) must be considered with caution and would likely need orthogonal experiments to support. Derived from the structural model, the authors validate important sites using

mutagenesis in vitro and in vivo. Overall, the study is of high quality but several questions remain. In the context of the literature the stoichiometry of the interactions was unclear and while the structure suggests a specific stoichiometry and nucleotide state, the biochemistry leading up to the cryo-EM samples is lacking and requires more stringent characterization to demonstrate the size/composition of the complex and the role of the nucleotides in the mechanism and how substrates are released. In my opinion, key experiments and questions remain that need to be addressed prior to publication in Nature Communications.

1. Lines 42-44. I don't understand the distinction between polypeptide and peptide in these two sentences and coupled with the mention of changes in association and dissociation rates makes the description confusing. The key concept here is that DnaK in the ATP bound state has low affinity for substrate and in the ADP has a high affinity. This description needs to be simplified.
2. Overall, it would be really useful to have an initial biochemical characterization of the complexes to confirm the stoichiometry of the assembly using SEC or SEC-MALS (or other methods) prior to cryo-EM analysis to ensure correct interpretation of the components. This is particularly important as the stoichiometry of DnaK:GrpE appears to be unclear in the literature – this would allow the authors to make stronger claims regarding the mechanism.
3. Purified DnaK (and Hsp70s generally) often copurify with ATP. Have the authors checked the nucleotide state of DnaK in the cryo-EM sample preparations?
4. While the ITC binding data and the 2D class averages for DnaK:GrpE in the absence of nucleotide are consistent and show that binding between DnaK and GrpE is tighter in the absence of nucleotide it would be really nice to have orthogonal validation of this observation using ITC.
5. Based on the proposed model, the prediction is that DnaK(ADP) should bind to GrpE releasing ADP, while a preformed DnaK:GrpE complex would dissociate upon addition of ATP. In the cryo-EM or ITC conditions this is really not tested. For example, in the supplementary Fig. 1c-d, for the DnaK+DnaJ2+GrpE+peptide or DnaK+DnaJ2+GrpE+inhibitor conditions was ATP included? In an ideal scenario, DnaK would be preincubated with ATP+Mg followed by addition of substrate preincubated with DnaJ2 followed by GrpE. As clearly highlighted by the authors model the order of events is really important and mixing all the components at once does not allow control of the reaction.
6. The title is catchy but perhaps misleading as the data in the paper suggest there is dynamicism in the DnaK:GrpE complex but there is no direct evidence that the dynamicism leads to concomitant release of the substrate from the SBD. I would recommend softening the title.
7. Mg is really important for any reactions with ATP and DnaK, its not clear whether this was included in the structural studies?
8. For the analysis of the GrpE mutants, it would be good to show SEC profiles and CD curves to confirm that each mutant is properly folded and the loss of binding is not due to misfolding/aggregation. Are all the GrpE mutants still dimers?
9. Derived from the structural model of DnaK:GrpE – have the authors tested the role of GrpE dimerization on binding to DnaK?
10. In vivo and in vitro validation is incredibly important and a key aspect of this work. To capture DnaJ2/substrate into the complex, have the authors tried glutaraldehyde cross-linking, with/without ATP/substrate to capture different states of the interactions in vitro and even possibly in cells?

Minor comments

1. Report ITC binding experiments as averages with standard deviation across the replicates.

2. Typo in line 988. "angel" should be "angle"

Reviewer #1 (Remarks to the Author):

DnaK is the eubacterial version of the molecular chaperone Hsp70. The protein consists of a nucleotide binding domain (NBD) and a substrate binding domain (SBD), whose nucleotide state and binding properties are allosterically coupled. DnaK binds to folding intermediates and misfolded protein species, and supports their folding in an ATP hydrolysis driven conformational cycle, which is regulated by the cochaperones DnaJ and GrpE. DnaJ recruits DnaK-ATP to substrates and triggers ATP hydrolysis, which leads to tight substrate binding. GrpE binding to DnaK triggers release of ADP, allowing binding of ATP and substrate release. The previous crystal structure of the *E. coli* GrpE-NBD complex was asymmetric and showed how one subunit in the GrpE dimer wedged into the nucleotide binding cleft. The coiled-coil domain in GrpE contributed to NBD binding and a preceding flexible part to substrate release. In contrast to the observed 2:1 stoichiometry, a later crystal structure of a *Geobacillus kaustophilus* GrpE-DnaK complex suggested a 2:2 stoichiometry, or 4:4 if DnaK-DnaK pseudo-substrate interactions in the crystal lattice are considered.

The present manuscript now reports the cryoEM structure of a complex of DnaK and GrpE from *M. tuberculosis* at 3.7 Å resolution. The complex has 2:1 stoichiometry. The interaction with the NBD seems to resemble the interaction observed in the *E. coli* complex (Here, a structure comparison by superposition and the r.m.s.d. value would be informative.). In addition, an interaction of the SBD with the coiled-coil helix pair of GrpE is observed, apparently similar to the interaction with one of the DnaK copies in the structure of a *Geobacillus kaustophilus* GrpE-DnaK complex (Again a structure comparison like above would be informative.). The C-terminal part of the SBD seems poorly ordered/defined in the density. Whether a substrate or a substrate-mimicking inhibitor is bound or not is unclear. Presence or absence of this ligand in the complex with DnaK should be demonstrated independently.

Thanks for the good suggestions. We have added a new figure (**Supplementary Fig. 7**) to show the superimposition of the DnaK-GrpE complexes of *Mtb*, *E. coli*, and *G. kaustophilus*. The r.m.s.d. values are 3.9 Å between *Mtb* and *Ec* and 5.3 Å between *Mtb* and *Gk*. We have also provided the r.m.s.d. values for individual subunits.

Per reviewer's suggestion, we have now examined the nucleotide and substrate binding in the *Mtb* DnaK-GrpE complex. We preincubated DnaK with or without ATP or ADP and then added DnaJ2 that was preincubated with a substrate peptide. After 30 min incubation, GrpE was added to the reaction mixture. We next removed the unbound peptides and nucleotides by gel filtration and collected the peak fractions corresponding to the DnaK-GrpE complex for mass spectrometry. The result showed that the protein complex does not contain nucleotides (ATP or ADP) or substrate peptides. This result is consistent with our cryo-EM observation and support the conclusion that GrpE binding facilitates the release of both nucleotide and substrate by DnaK. We have added these data in the main text and in **Supplementary Fig. 4**.

Next, the authors use site-directed mutagenesis to probe the observed contact interfaces. Surprisingly, mutation of the contacts in the helix pair region in GrpE seemed to affect the affinity and nucleotide exchange activity more than mutations in the wedge and the 4-helix bundle, closer to the nucleotide binding cleft. In vivo, the former mutation did not support growth, in contrast to the latter. So, the contacts to the NBD and SBD close to the DnaK interdomain linker seem to contribute substantially to the GrpE binding affinity.

When mutations were introduced in the interface regions in DnaK, mutations in subdomain IIB (next to the wedge) had the strongest effect on affinity. This mutation disrupted DnaK function in vivo. Mutation of residues contacting the helix pair had GrpE affinity similar to wildtype, but were non-functional in vivo. Here it is not clear whether these mutations disrupt DnaK function by other means than interactions with GrpE, for example by disrupting the allosteric coupling between the DnaK domains or

the interaction with DnaJ.

We thank the reviewer for raising the good point. We have revised the main text to add a caveat: “We suggest that the lethal DnaK mutations (*Mut-3* and *Mut-4*) that had no influence on GrpE binding may affect the DnaK allosteric coupling between NBD and SBD or affect DnaK interaction with DnaJ1/DanJ2”, as the reviewer advises.

Finally, the authors study the dynamics of the complex by eigenvector analysis. The authors propose that the wobbling of the domains in the complex facilitates nucleotide and substrate release. This seems plausible to me. However, I would not call this “allostery”. That the DnaK nucleotide binding cleft engages in alternative contacts in the complex with GrpE, which weaken nucleotide binding, I would describe as allostery. But the alpha-helical lid over the substrate binding pocket is modelled in place for substrate interactions unlike in the DnaK-ATP complex. If MtGrpE binding is substantially weakening the interaction of MtDnaK with substrate, an alternative mechanism seems to be required, for example displacement by a substrate-mimicking peptide in the disordered N-terminal tail. This could be investigated by an N-terminal truncation mutant of MtGrpE. To show that substrate release from DnaK is triggered by GrpE binding or that its affinity is lowered in the complex, a biochemical assay should be set up.

We thank the reviewer for the insightful suggestions. We have removed “allostery” from the title and revised it to “Structure of the *M. tuberculosis* DnaK-GrpE complex reveals key features that controls DnaK’s nucleotide exchange and substrate release”.

We appreciate the reviewer’s brilliant idea that the long GrpE N-tail may displace substrate peptide in the DnaK SBD. We have designed a GrpE truncation (aa 43-235) by removing the disordered 42-residue N-tail. We produced the truncated GrpE protein and performed a fluorescence polarization (FP) experiment using a fluorescently tagged NR (F-NR) with an increasing amount of GrpE WT and its truncation mutants. We first preincubated DnaK with ADP and F-NR before adding WT or mutant GrpE at an increasing concentration. We found that WT GrpE effectively competes with F-NR for DnaK with an effective half-maximal concentration of 16.3 μ M, suggesting that GrpE binding triggers substrate release from DnaK (revised **Fig. 1H**). As anticipated by the reviewer, GrpE_ Δ N-tail indeed blocked substrate release from DnaK, suggesting that the GrpE N-tail plays an important role in triggering substrate release from DnaK. This observation suggests that the Mtb GrpE allosterically regulate DnaK via the disordered N-tail.

I am not sure if the present structure goes much beyond the previously existing structures. The more interesting parts - the DnaK SBD and its helical bundle and the beta-domain in the second GrpE copy - are rather ill-defined in the present density. The mutational analysis seems to be contradictory in parts. Experimental evidence for substrate release from DnaK via GrpE binding in *M. tuberculosis* is missing.

We appreciate the reviewer’s constructive critiques and thoughtful suggestions. We acknowledge the partially mobile DnaK SBD and the distal beta-sheet of the second GrpE may reflect the dynamic nature of the system, as cryo-EM observes the solution poses rather than a pose locked in a lattice in crystal structures. We have performed suggested experiment to show GrpE stimulated substrate release from DnaK. We thank the reviewer for helping us to strengthen our work.

Detailed points:

How much of the N-terminal regions of GrpE could not be modelled in the complex? In case large portions are missing in the model, this should be mentioned in the main text. According to the PDB validation report also large C-terminal portions were not modelled in one of the GrpE subunits.

The modeled region is aa 44 -187 in GrpE A and aa 45 -134 in GrpE B. We have included this information in the revised text (**Fig. 2A**).

p. 3, line 86. What happens when DnaK-DnaJ2-GrpE are mixed in 2:2:2 ratio? Is only one copy of DnaK incorporated into the complex?

To address reviewer's comment, we have conducted a new experiment by mixing 10 μ M DnaK with 10 μ M DnaJ2 that was preincubated with the substrate peptide and incubating the mixture for 30 min. We then added 10 μ M GrpE to the reaction mixture, incubated for additional 10 min, and performed a SEC-MALS experiment. The result shows a molecular mass of \sim 114 kDa, corresponding to the 115-kDa complex observed in our cryo-EM studies. Therefore, at the mixing ratio of 2:2:2, the assembled complex is composed of one copy of DnaK and two copies of GrpE. We have included the new data in the revised **Fig. 1E**. In addition to the SEC-MALS, we also examined the reaction mixture by cryo-EM. 2D class averages show similar features as observed in the previously assembled samples, further demonstrating that the adding more DnaK does not change the 1:2 ratio of DnaK and GrpE in the assembled complex (**Fig. 1E-G**).

p. 4, line 119. Normally the lobes in DnaK are designated I and II and the respective subdomains A and B (and not vice versa).

We apologize for this inconsistency and have corrected in revision.

Fig. 3 There are several contacts shown which are rather unlikely at neutral pH, for example hydrogen bonds between D362 and E57, or between R396 and R75. Can you please show how well the respective sidechains are defined in density? If they are not defined well, other side chain conformations seem more likely. Is the sequence register in the GrpE coiled-coil helices unambiguously clear?

We thank the reviewer for catching the error, and we have revised the H-bond assignment. We have removed the H-bonds that are geometrically unfavorable (T60-V150, T60-A149) or unlikely to form (D362-E57, R396-R75) at pH 8.0 used in our experiment.

We now show superimpositions of the atomic model and EM map in selected regions in the DnaK NBD, SBD, GrpE subunit A and subunit B (**Supplementary Fig. 3A-C**) and show the EM densities for the interacting residues in contact regions I-III (**Supplementary Fig. 3D-G**). Most of the sidechains of DnaK and GrpE are well defined in our EM map except for the DnaK SBD which we have clearly noted to be of lower resolution.

Also, for some of the hydrogen bonds shown, the geometries seem unfavorable (for example from T60). The hydroxyl group of Y257 seems to engage in H-bonds to three carbonyl groups, exceeding its hydrogen bonding capability.

We thank the reviewer for pointing it out. As mentioned above, we have reexamined and reassigned H-bonds and updated the interactions in **Fig. 3**.

p.5, line 130 The references to the "lower left side" and "lower right side" do not work without a specific Figure for illustration. Better use a description independent of a specific view: The disjointed SBD domains exhibit diverse orientations with respect to the NBD in the available structures?

We thank the reviewer for the suggestion and have changed the description accordingly.

Fig. 4B The luciferase folding yield (\sim 12%) appears very poor. Either the Mtb chaperones do not work well with this substrate, or the concentrations (or concentration ratios) of the proteins in the assay are poorly chosen. In the latter case, a clearer result for GrpE Mut-2 and Mut-3 might be obtained by concentration optimization.

Reactivation of heat-denatured luciferase by Mtb DnaK with DnaJ and GrpE has been reported previously by Lupoli et al (PMID: 27872278). The optimized system recovers a maximum of 15-20% percent, and addition of ClpB can increase the recovery somewhat. Mtb has two J proteins (DnaJ1 and DnaJ2), which coordinate in the reactivation of heat-treated substrate. We used the concentrations and conditions that have been optimized in the Lupoli paper. The relatively poor activity may be due to the lack of important factors such as DnaJ1 and ClpB.

We have repeated the reactivation experiments by including the new mutant (Mut-4). We observed a similar activity (revised **Fig. 4B**): the GrpE Mut-1 and Mut-4 had a severe effect on the luciferase refolding activity of DnaK, while GrpE Mut-2 and 3 had little impact. We have also repeated the ADP/ATP exchange assay by including the new GrpE Mut-4 (revised **Fig. 4C**).

Fig. 4D, panel ii) What was the approximate K_D ? The curve does appear interpretable.

We have now included the K_D value of the GrpE Mut-1 (1.6 μ M; **Fig. 4D-ii**).

Fig. 6A There are multiple structures of the E. coli DnaK in the PDB. It is not clear which one was chosen in subpanel i – please specify in the Figure legend.

We have added the PDB IDs for both DnaK (1DKG) and GrpE homologues in the revised figure legend. We have also added the accession numbers of the DnaK homologues used in the sequence alignment.

In the GrpE-DnaK-NBD complex, the conformational change was described as a 14° rotation of subdomain IIB with respect to the Hsc70 NBD (ADP complex, I presume) in the first structure paper (Harrison et al., 1997). So, is there an additional outward rotation of subdomain IB in the M. tuberculosis complex? Please describe this more clearly in the main text.

Yes, NBD lobe I of Mtb DnaK is slightly more rotated against lobe II, than that of *E coli* NBD. We have updated Fig. 6A-B by aligning all structures based on Lobe II (subdomains IIA & IIB). We have now included the alignment details in revised text: *“We further compared the structures of the GrpE-bound Mtb DnaK with that of ADP-bound, ATP-bound and nucleotide-free DnaK structures, and found that GrpE induces ~13° rotation of DnaK IB and 6° rotation of DnaK IIB relative to the ADP-bound DnaK, and ~17° rotation of DnaK IB, and 5° rotation of DnaK IIB relative to the ATP-bound DnaK, while compared to the nucleotide-free DnaK, binding of GrpE induces a 10° rotation of DnaK IB and a 5° rotation of DnaK IIB.”*

The arrows in subpanel ii seem to point into the wrong direction.

We have corrected the arrows in new Figure 6 subpanel v.

Fig. 6B This superposition is pretty messy. The conformations cannot be distinguished easily. Better compare one by one. Was there a single subdomain in all structures held in place? To which structure pair does the curved arrow in subpanel i refer?

We apologize for the lack of clarity. We now show the aligned structures individually in revised **Fig. 6B**. We aligned the structures in ChimeraX based on lobe II (subdomains IIA & IIB). Thus, a large rotation of subdomain IB with respect to subdomain IIB was observed. In the revised figure, we have updated the curved arrows and added the rotation angles, which indicate the movement of the lobe I and II in comparison to NBD of Mtb DnaK.

In the ATP- and ADP-bound states, the nucleotide binding pocket should be closed. Why is there an opening angle?

We have updated the figure by removing the confusing angles. We have labeled the ADP-bound GkDnaK-NBD and ATP-bound EcDnaK-NBD as the closed state, and MtbDnaK-NBD and MgDnaK-NBD as the open state.

In the ATP state, the NBD lobes of Hsp70 are known to exhibit sideways shearing compared to the ADP state. So, there are rather complex conformational changes between the states, which cannot be adequately described with one angle.

For readers not familiar with Hsp70, it would be useful to include the nucleotide to show the position of the nucleotide binding pocket.

We agree and have removed the angles. We have now included the bound nucleotide in the structures of GkDnaK-NBD and EcDnaK-NBD (revised **Fig. 6B**).

Reviewer #2 (Remarks to the Author):

The manuscript by Xiao, et al. describes their work using cryo-EM and functional assays to understand how the Mtb GrpE helps DnaK to function in ADP and refolded substrate release. They found that the GrpE long N-terminal α -helix plays a dual role in the asymmetric binding to DnaK and the allosteric control of DnaK's NBD and SBD domains. They further found that the DnaK-GrpE complex is highly dynamic and undergoes coordinated motions that are unique both scientifically and technically. Although there is prior work on the GrpE complex with DnaK NBD, this work provides complete and dynamic views of how GrpE regulates DnaK's structure and function. The study represents an excellent example of showing the utility of cryo-EM to explore the structure and dynamics of protein complexes. Functional data support their structural analysis. The manuscript is well written although there are typos and confusing statements. Below are my comments for the authors to consider.

We thank the reviewer for recognizing the importance of our work and the many thoughtful suggestions.

Scientific: The allosteric regulation of NBD and SBD by ATP binding and substrate release has been well-studied both structurally and functionally. The regulatory role of GrpE in accelerating ADP release after ATP hydrolysis in DnaK NBD is also well known. Using the full-length DnaK, the authors determined the GrpE-DnaK complex structure and propose that the GrpE dimer allosterically regulates the ADP release and substrate refolding. The solved complex structure does not contain ADP or refolded substrate. So, it is not clear to the reviewer if GrpE-mediated ADP release may regulate refolded substrate release and vice versa. Is it possible that the release of the refolded substrate may be triggered by ATP hydrolysis, but not ADP release, and thus prior to the GrpE binding? The authors should discuss other possible scenarios in the discussion session.

Thanks for raising the interesting point. To demonstrate the GrpE-mediated substrate release, we have now fluorescently tagged the substrate NR (F-NR) and performed fluorescence polarization (FP) experiment with DnaK. We show that F-NR binds Mtb DnaK with a high affinity and a slow binding kinetics in the presence of ADP or in the absence of a nucleotide (revised **Supplementary Fig. 5**). We next performed competition assay with increasing concentrations of GrpE. Briefly, we preincubated DnaK with ADP and F-NR before adding to the reaction mixture an increasing amount of GrpE. From the fitting result, we found that GrpE competes with F-NR for binding to DnaK. The estimated effective half-maximal concentration suggests that GrpE binding triggers substrate release from DnaK (revised **Fig. 1H**).

Technical: Cryo-EM is the ideal tool to study the dynamic structures of protein complexes. The GrpE-DnaK has a molecular mass of only about 100 kDa, which is relatively small for cryo-EM analysis. I have several technical comments concerning the cryo-EM analysis of this small complex.

1) The complex structure was reconstructed at 3.7 Å resolution. However, in the manuscript, there is a

lack of validation of the reconstructed map. It's suggested that the authors provide FSC curves for masked and unmasked maps as well as a local-resolution map to show the quality of their reconstruction. In Figure 3, the authors show residue-residue interactions between GrpE and DnaK for the three contracts. However, there are no cryo-EM densities to support any of these interactions. The authors should provide cryo-EM densities or explanations on how they validate all or some of the interactions.

Thanks for the good suggestion. We have now included the FSC curves and local resolution maps in the revised **Supplementary Fig. 2**. We have also added a new **Supplementary Fig. 3** to show the EM densities for DnaK NBD, DnaK SBD, and GrpE Mol A and Mol B, as well as zoomed views of the EM map at contact regions I-III.

2) The Mtb DnaK SBD contains only 256 residues (from Supplementary Figure 4). The authors used the focused refinement to get a reconstruction of the DnaK-SBD at 5.8 Å resolution. The authors should validate their cryo-EM map by using FSC curves. In addition, they need to show the map quality and the fitness of the atomic model they built. The map quality for the SBD also relates to the quality of the model they used to draw GrpE-SBD interactions in Figure 3E. In addition, with a domain of 256aa, its projected signal is very weak. I wonder how strong the signal they can get after partial signal subtraction of other components. The authors should compare the maps before and after focused refinement to see if any improvement in the map quality.

We thank the reviewer for the comment. Relatively small features have been visualized via signal subtraction and focused refinement in the literature. Our focused refinement of the 28-kDa DnaK-SBD resolved two-thirds of the domain, but the remaining one-third that contains the α -helical lid region (403-442) was still missing. We have now included a superimposition of the atomic model and EM map of DnaK-SBD (revised **Supplementary Fig. 3B**). The model for the unresolved lid region was derived by rigid body docking of the published SBD structure. We should note that the GrpE-interacting region that we discussed in the main text is in the resolved two-third region of the SBD with solid EM density (revised **Supplementary Fig. 3F**). We have added a comparison of the DnaK-SBD EM maps before and after focused refinement (revised **Supplementary Fig. 2**) and the superimposition of model and EM map at the contact regions I-III (**Supplementary Fig. 3E-H**).

3) The authors used multibody refinement and principal component analysis to derive motions in the GrpE-DnaK system. The total molecular mass of the system is about 100 kDa. The authors split the system into four bodies. The same to my concern above, the authors should validate their refined maps to see if they obtained better maps from the multi-body refinement. It's a question to me if RELION can perform reliable four-body refinements with individual bodies less than 100 kDa. The noise associated with partial signal subtraction makes the multibody refinement very challenging. Instead, have authors tried 3D variability analysis in cryoSPARC?

We thank the reviewer for the comment. We have now compared the maps before and after multi-body refinement in RELION and found the map quality was not markedly improved by multi-body refinement (**revised Supplementary Fig. 2**). We have also performed 3DVA in cryoSPARC and found similar domain motions in the complex. We chose to present the RELION-detected motions in the manuscript as these maps appear to have slightly better quality.

Specific comments:

1) The authors repeatedly use DnaK/Hsp70 in the introduction. They can mention Hsp70 once and use DnaK in the rest of the introduction.

We have revised accordingly.

2) Line 51: What is NEF? Please spell out the full name.

We have added the definition of NEF in line 36.

3) Lines 73-74: "This property contrasts with DnaK of the other bacteria that is nonessential." Does this mean that DnaK is dispensable in other bacterial species including *M. smegmatis* that was used in this study? Please cite a reference to support this statement.

We apologize for the confusing statement. DnaK is not essential in normal growth conditions for *E. coli* but is essential for mycobacteria, which include *M. smegmatis*. We have revised as "*DnaK is nonessential in normal growth conditions in E. coli but is essential in mycobacteria for cell growth and protein folding*".

4) Lines 139-142: Are the contact region III interface between GrpE B and DnaK SBD also charge-charge interactions? How was the contact area of the region III interface (891 Å²) calculated, from the atomic model or cryo-EM map?

The interface between GrpE B and DnaK SBD in contact region III is mediated by a charge-charge interaction (D451-R64) (revised **Fig. 3E**). Please note the contact region is well within the resolved region of the SBD in the EM map, and the contact area is calculated from the atomic model in ChimeraX. This has been added in the revised Method.

5) Lines 157-164: What is the experimental evidence to support the interactions shown in Figure 3? Without showing cryo-EM densities, it's hard to evaluate the quality of the specific interactions described in the section from lines 157 to 164.

We have now included map-model superimposition in several key regions including DnaK NBD, SBD, and GrpE subunits A and B (revised **Supplementary Fig. 3A-C**). Most sidechains in DnaK and GrpE are well defined in the EM map (with the exception in DnaK SBD). Superimpositions of the model and map are also shown for contact regions I-III (**Supplementary Fig. 3D-G**).

6) Line 178: did the authors add Mg²⁺ which is required for the ATP hydrolysis activity? If they did, they should include Mg²⁺ and its concentration here.

All in vitro reactions were conducted in buffer containing 20 mM MgCl₂. This information was included in the Materials and Methods section (line 560) but may have been missed.

7) Line 210: "Furthermore, neither Mtb GrpE Mut-2 nor Mut-3 allele supported viability, demonstrating that these mutant proteins could complement for essential...". This sentence is confusing and contradicts the next sentence "However, GrpE Mut-1 (Fig. 4F) or GrpE Mut1-3 (data not shown) did not support viability, indicating that the Mut-1 residues are essential for DnaK system.". It reads that Mut-2 and Mut-3 are lethal. Please revise the sentence for consistency.

We apologize for the confusing statement. We have revised the main text as "*Furthermore, both Mtb GrpE Mut-2 and Mut-3 alleles supported viability, demonstrating that these mutant proteins could complement for essential GrpE function in M. smegmatis*".

8) Line 212: "(data not shown)". I think the journal policy requires showing the data which could be presented as part of the supplementary data.

We apologize for the omission. We have added the data in the revised **Supplementary Fig. 8**.

9) Line 216: It seems that DnaK is essential for *M. smegmatis* viability. This statement is confusing with the authors' argument of DnaK as a drug target because it's vital only to Mtb. The authors should revise their text to address such discrepancies.

We have addressed this issue in the response to the reviewer's comment #3.

10) Line 260: If Mut-1 is insoluble in vitro (Line 246), the authors should explain why it supported bacteria growth in vivo.

Thanks for raising this important point, which has forced us to try harder for this mutant. We are happy to report we are now able to produce soluble DnaK Mut-1 protein after optimizing the expression conditions. We found DnaK Mut-1 has severely reduced reactivation activity for denatured luciferase. The relevant figure has been updated to include this mutation (**Fig. 5**).

11) Lines 297-298: There are many factors contributing to a low or modest resolution of a cryo-EM map. What about air-water interface issue? The authors should include a local-resolution map and use it to suggest the presence of conformational heterogeneity.

We thank the reviewer for the comment. We have now included the local resolution map and the particle distribution map in the revised manuscript (**Supplementary Fig. 2E-F**). We agree with the reviewer that the primary factor limiting the achievable resolution in our sample is conformational heterogeneity, which is revealed in our multi-body refinement result (**Fig. 7**).

12) Lines 333-334: The authors claim that Arg64 and Arg75 contribute to the interactions with both DnaK NBD and SBD. Their GrpE Mut-1 contains six mutations (R64A, N71A, R73A, K74A, R75A and R78A). Did they try to make an R64A-R75A double mutant to evaluate their importance in ADP release and/or substrate release?

We thank the reviewer for the insightful suggestion. We have constructed the R64A-R75A double mutant (Mut-4), purified the mutant protein, and assayed it for luciferase reactivation and ADP/ATP exchange. Like Mut-1, Mut-4 greatly reduced the reactivation activity and severely affected the ADP/ATP exchange rate, indicating that interactions mediated by GrpE R64 and R75 are critical to the DnaK-GrpE function (revised **Fig. 4B-C**).

13) Supplementary Figure 2. Missing a scale bar in the raw micrograph. "right corner" should be "left color". Eulerian angel distribution" should read as "Eulerian angular distribution".

Thanks. We have added the scale bar in the raw micrograph and corrected the typos in legend.

14) Supplementary Figures 3& 4: Missing accession numbers for the GrpE and DnaK homologs used in their multiple sequence alignments.

The accession numbers of the GrpE and DnaK homologs have been added in the legends of revised **Supplementary Figs. 6 & 9**.

Reviewer #3 (Remarks to the Author):

In this study by Xiao et al, the authors use FL DnaK and GrpE from *M. Tub.* to understand the mechanism of binding/allostery that promotes ADP release from DnaK and thus facilitating nucleotide exchange required for rounds of substrate refolding by DnaK. The authors reconstitute the complex from recombinant components of DnaK:GrpE and use cryo-EM to determine a structure uncovering a 1:2 stoichiometry. The cryo-EM data is of reasonable quality but require additional processing and masking to interpret the dynamics and given that the resolution of the maps is not fantastic the

conclusions drawn about the dynamics (and small changes in angular deviation) must be considered with caution and would likely need orthogonal experiments to support. Derived from the structural model, the authors validate important sites using mutagenesis in vitro and in vivo. Overall, the study is of high quality but several questions remain. In the context of the literature the stoichiometry of the interactions was unclear and while the structure suggests a specific stoichiometry and nucleotide state, the biochemistry leading up to the cryo-EM samples is lacking and requires more stringent characterization to demonstrate the size/composition of the complex and the role of the nucleotides in the mechanism and how substrates are released. In my opinion, key experiments and questions remain that need to be addressed prior to publication in Nature Communications.

We are grateful to the reviewer's insightful comments. We have addressed all their questions by performing suggested experiments, improving data analysis and presentation, and revising the main text.

1. Lines 42-44. I don't understand the distinction between polypeptide and peptide in these two sentences and coupled with the mention of changes in association and dissociation rates makes the description confusing. The key concept here is that DnaK in the ATP bound state has low affinity for substrate and in the ADP has a high affinity. This description needs to be simplified.

We apologize for the confusing sentence and have rephrased: "*In the ATP-bound state, DnaK's affinity for substrate is low but substrate association and disassociation rates are high. Upon ATP hydrolysis, affinity for substrate increases 10- to 50-fold, but substrate association and dissociation rates decrease 100- and 1000-fold, respectively.*"

2. Overall, it would be really useful to have an initial biochemical characterization of the complexes to confirm the stoichiometry of the assembly using SEC or SEC-MALS (or other methods) prior to cryo-EM analysis to ensure correct interpretation of the components. This is particularly important as the stoichiometry of DnaK:GrpE appears to be unclear in the literature – this would allow the authors to make stronger claims regarding the mechanism.

Following the reviewer's advice, we have now characterized the stoichiometry of the DnaK-GrpE assembly using SEC-MALS. The measured mass of the assembled complex is consistent with a 1:2 stoichiometry of DnaK:GrpE (**Fig. 1E**). This solution result is also consistent with the 1:2 complex captured by our cryo-EM analysis.

3. Purified DnaK (and Hsp70s generally) often copurify with ATP. Have the authors checked the nucleotide state of DnaK in the cryo-EM sample preparations?

Per reviewer's suggestion, we have examined the presence of ATP in the purified DnaK sample by mass spectrometry. The result confirms the reviewer's impression and shows that purified DnaK indeed contains ATP. We next examined the nucleotide state of DnaK-GrpE complex used for cryo-EM studies. We first incubated DnaK with or without ATP or ADP, then added DnaJ2 preincubated with the substrate peptide and incubated the mixture for 30 min, and finally added GrpE and incubated for another 10 min. We then removed unbound peptides and nucleotides by gel filtration. The peak fractions corresponding to DnaK-GrpE complex were collected and sent for mass spectrometry measurement. The result showed the DnaK-GrpE complex does not contain any nucleotide (ATP or ADP) or substrate peptide. This result supports our structure-based conclusion that GrpE facilitates the release of both nucleotide and substrate by DnaK. We have added these findings into the main text and in **Supplementary Fig. 4**.

4. While the ITC binding data and the 2D class averages for DnaK:GrpE in the absence of nucleotide are consistent and show that binding between DnaK and GrpE is tighter in the absence of nucleotide it would be really nice to have orthogonal validation of this observation using ITC.

Following the reviewer's advice, we have performed ITC experiment for DnaK and GrpE in the presence of ADP or ATP. The result show that the binding affinity of DnaK and GrpE in the presence of ADP is similar to the binding affinity in the absence of ADP, while in the presence of ATP, the binding affinity is 10-15 times lower than that in the presence or absence of ADP. We have added these findings into the main text (revised **Fig. 7B**).

5. Based on the proposed model, the prediction is that DnaK(ADP) should bind to GrpE releasing ADP, while a preformed DnaK:GrpE complex would dissociate upon addition of ATP. In the cryo-EM or ITC conditions this is really not tested. For example, in the supplementary Fig. 1c-d, for the DnaK+DnaJ2+GrpE+peptide or DnaK+DnaJ2+GrpE+inhibitor conditions was ATP included? In an ideal scenario, DnaK would be preincubated with ATP+Mg followed by addition of substrate preincubated with DnaJ2 followed by GrpE. As clearly highlighted by the authors model the order of events is really important and mixing all the components at once does not allow control of the reaction.

Per reviewer's suggestion, we have now preincubated DnaK with MgCl₂ and ATP (sample 1), ADP (sample 2), or no nucleotide (sample 3), followed by the addition of pre-incubated substrate peptide and DnaJ2, and after 30 min incubation, finally added GrpE. We examined the three samples by cryo-EM. 2D averages show the presence of the 1:2 DnaK-GrpE complex in all three samples. We have included this observation in the revised text (**Supplementary Fig. 1E-G**).

6. The title is catchy but perhaps misleading as the data in the paper suggest there is dynamicism in the DnaK:GrpE complex but there is no direct evidence that the dynamicism leads to concomitant release of the substrate from the SBD. I would recommend softening the title.

We appreciate the advice and have revised the title to "Structure of the *M. tuberculosis* DnaK-GrpE complex reveals key features that control DnaK's nucleotide exchange and substrate release".

7. Mg is really important for any reactions with ATP and DnaK, its not clear whether this was included in the structural studies?

For structural studies, the DnaK-GrpE complex was assembled in buffer containing 10 mM MgCl₂, and all biochemical studies were conducted in buffer containing 20 mM MgCl₂. We mentioned this in the materials and methods (line 466 and line 543) but may have been missed.

8. For the analysis of the GrpE mutants, it would be good to show SEC profiles and CD curves to confirm that each mutant is properly folded and the loss of binding is not due to misfolding/aggregation. Are all the GrpE mutants still dimers?

Thanks for the good suggestion. We have now included the SEC profiles of all GrpE mutants in **Supplementary Fig. 11**. The mutant GrpE proteins exist as a dimer in solution as indicated by an arrowhead, similar to the WT GrpE. We collected the fractions corresponding to the well-folded GrpE dimer for all biochemical studies.

9. Derived from the structural model of DnaK:GrpE – have the authors tested the role of GrpE dimerization on binding to DnaK?

Because the dimerization interface is so extensive, even the mutant GrpE proteins were predominantly dimeric in solution. So, we did not have purified monomeric GrpE to examine this issue. But it is well established the GrpE exists in solution and functions as a dimer.

10. In vivo and in vitro validation is incredibly important and a key aspect of this work. To capture DnaJ2/substrate into the complex, have the authors tried glutaraldehyde cross-linking, with/without ATP/substrate to capture different states of the interactions in vitro and even possibly in cells?

Thanks for the comment. In fact, capturing the ternary DnaK-GrpE-DnaJ was our original goal. We have tried numerous times to capture the complex of DnaK-GrpE with DnaJ2/substrate. However, we have only seen the binary complex of DnaK-GrpE (**Supplementary Fig. 1E-G**). We have also tried glutaraldehyde cross-linking and with or without ATP/substrate. But we have only observed the DnaK-GrpE complex (**Supplementary Fig. 1H**). In our response to the reviewer's comment #3, we have shown with mass spectrometry that a stable DnaK-GrpE complex contains neither nucleotide nor peptide substrate.

Minor comments

1. Report ITC binding experiments as averages with standard deviation across the replicates.

We have added the averages with standard deviation of the ITC experiments across the replicates in **Supplementary Table 5**.

2. Typo in line 988. "angel" should be "angle"

Corrected.

Reviewer #1 (Remarks to the Author):

My points of criticism have been adequately addressed in the revised version of the manuscript. The present work is an important addition to the molecular chaperone field.

The statement in line 397, that the beta-sandwich domain of Hsp110 wedges open the NBD of Hsp70 is however incorrect. In the structures of the complex there were no contacts observed between this domain and the NBD. It appears rather that contacts between the helical bundle domain of Hsp110 and subdomain IIB of Hsp70 stabilize the NBD in an "open nucleotide-binding cleft" conformation.

Minor point:

line 302. Replace DanJ2 with DnaJ2.

Reviewer #2 (Remarks to the Author):

The quality of the revised manuscript has been greatly improved. All my concerns were mostly addressed satisfactorily. The manuscript should be published in Nature Communications.

Reviewer #3 (Remarks to the Author):

All my comments have been addressed in the revised manuscript.

Reviewer #1 (Remarks to the Author):

My points of criticism have been adequately addressed in the revised version of the manuscript. The present work is an important addition to the molecular chaperone field.

We thank the reviewer for recognizing the importance of our work and the many thoughtful suggestions.

The statement in line 397, that the beta-sandwich domain of Hsp110 wedges open the NBD of Hsp70 is however incorrect. In the structures of the complex there were no contacts observed between this domain and the NBD. It appears rather that contacts between the helical bundle domain of Hsp110 and subdomain IIB of Hsp70 stabilize the NBD in an "open nucleotide-binding cleft" conformation.

We thank the reviewer for catching the error, and we have revised as advised:

"This conformational switch resembles those found in the mammalian Hsc70 ATPase and their bacterial homolog DnaK ATPase upon binding to their NEFs (Bag in mammals and GrpE in bacteria), while in the yeast Hsp70 ATPase, the contacts between the helical bundle domain of Hsp110 and subdomain IIB of Hsp70 stabilize the NBD in an open nucleotide-binding cleft" conformation".

Minor point:

line 302. Replace DanJ2 with DnaJ2.
Corrected.

Reviewer #2 (Remarks to the Author):

The quality of the revised manuscript has been greatly improved. All my concerns were mostly addressed satisfactorily. The manuscript should be published in Nature Communications.

We thank the reviewer for recognizing the importance of our work and the many thoughtful suggestions.

Reviewer #3 (Remarks to the Author):

All my comments have been addressed in the revised manuscript.

We are grateful to the reviewer's many insightful comments.